# Through the Looking Glass: Mirror Schrödinger Bridges

## Abstract

Resampling from a target measure whose density is unknown is a fundamental problem in mathematical statistics and machine learning. A setting that dominates the machine learning literature consists of learning a map from an easy-to-sample prior, such as the Gaussian distribution, to a target measure. Under this model, samples from the prior are pushed forward to generate a new sample on the target measure, which is often difficult to sample from directly. In this paper, we propose a new model for conditional resampling called *mirror Schrödinger bridges*. Our key observation is that solving the Schrödinger bridge problem between a distribution and itself provides a natural way to produce new samples from conditional distributions, giving in-distribution variations of an input data point. We show how to efficiently solve this largely overlooked version of the Schrödinger bridge problem. We prove that our proposed method leads to significant algorithmic simplifications over existing alternatives, in addition to providing control over in-distribution variation. Empirically, we demonstrate how these benefits can be leveraged to produce proximal samples in a number of application domains.

## 1 Introduction

Mapping one probability distribution to another is a central technique in mathematical statistics and machine learning. Myriad computational tools have been proposed for this critical yet often challenging task. Models and techniques for optimal transport provide one class of examples, where methods like the Hungarian algorithm (Kuhn, 1955) map one distribution to another with optimal cost. Adding entropic regularization to the static optimal transport problem yields efficient algorithms like Sinkhorn's method (Deming & Stephan, 1940; Sinkhorn, 1964), which have been widely adopted in machine learning since their introduction by Cuturi (2013). Static entropy-regularized optimal transportation is equivalent to a dynamical formulation known as the *Schrödinger bridge problem* (Schrödinger, 1932; Léonard, 2014), which has proven useful to efficiently compute an approximation of the optimal map paired with an interpolant between the input measures.

Inspired by these mathematical constructions and efficient optimization algorithms, several methods in machine learning rely on learning a map from one distribution to another. Beyond optimal transport, diffusion models, for instance, learn to reverse a diffusion process that maps data to a noisy prior. Special attention has been given to learning methods that accomplish this in a *stochastic* manner, i.e., modeling the forward noising process using a stochastic differential equation (SDE).

The most common learning applications of distribution mapping attempt to find a map from a simple prior distribution and a complex data distribution, either using a score-matching strategy (Song & Ermon, 2019; Ho et al., 2020; Song et al., 2021) or leveraging a formulation of the Schrödinger bridge problem (De Bortoli et al., 2021; Shi et al., 2022; 2023; Zhou et al., 2024); other learning applications map one complex data distribution to another (Cuturi, 2013; Courty et al., 2017).

In this paper, we focus instead on the understudied problem of mapping a probability distribution to *itself*, that is, finding a joint distribution whose marginals are both the same data distribution $\pi$. This task might seem inane at first glance, since two simple couplings satisfy our constraints: one is the independent coupling $p(x, y) = \pi(x)\pi(y)$, and the other is the "diagonal" map given by $p(x, y) = \pi(x)\delta_y$. The space of couplings between a measure and itself, however, is far richer than these two extremes and includes models whose conditional distributions are neither identical nor Dirac measures.

FIX

We focus on the class of self-maps obtained by entropy-regularized transport from a measure to itself. Formally, we define a *mirror Schrödinger bridge* to be the minimizer of the KL divergence $D_{\mathrm{KL}}(\mathbb{P} \parallel \mathbb{P}^0)$ over path measures $\mathbb{P}$ with both initial and final marginal distributions equal to $\pi$, where $\mathbb{P}^0$ is an Ornstein-Uhlenbeck process with noise $\sigma$. Mirror Schrödinger bridges are the stochastic counterpart to minimizing $D_{\mathrm{KL}}(p \parallel p^0)$, where $p^0$ is the probability density of the joint distribution associated with the path measure $\mathbb{P}^0$, over the joint distributions $p$ on $\mathbb{R}^n \times \mathbb{R}^n$ satisfying the linear constraints $\int p(x, y) dy = \pi(x)$ and $\int p(x, y) dx = \pi(y)$. While the former minimizes the Kullback-Leibler divergence on path space, the latter is a minimization over density couplings.

Despite its simplicity, this setting of the Schrödinger bridge problem suggests a rich application space. Couplings with the same marginal constraints have already proven useful to enhance model accuracy in vision and natural language processing by reinterpreting attention matrices as transport plans (Sander et al., 2022). Few works, however, consider this task from the perspective of optimizing over path measures or provide control over the entropy of the matching at test time. Albergo et al. (2023) propose a stochastic interpolant between a distribution and itself, but their interpolants are not minimal in the relative entropy sense and do not solve the Schrödinger bridge problem, even with optimization. Minimal interpolants in the relative entropy sense are those with minimal kinectic energy, and in applications, minimizing the kinectic energy of a path has been correlated to faster sampling (Shaul et al., 2023).

**Contributions.** We investigate the mirror Schrödinger bridge problem and demonstrate how it can be leveraged to obtain in-distribution variants of a given input sample. In particular, given a sample $x_0 \sim p_{\mathrm{data}}$, we build a stochastic process $\{\mathbf{X}_t\}_{t \in [0,1]}$ with minimal relative entropy under which the sample $x_0$ arrives at some $x_1 \sim p_{\mathrm{data}}$ with $x_1$ proximal but not identical to $x_0$.

Our contributions in this direction are twofold: first, on the theoretical side, we use the time symmetry of the mirror Schrödinger bridge to prove that it can be obtained as the limit of iterates produced via an alternating minimization procedure; and second, in applications, the implementation of our method allows for sampling from the conditional distribution $\mathbf{X}_1 \mid \mathbf{X}_0 = x_0$ in such a way that we can control how proximal a generated sample $x_1$ is relative to the input sample $x_0$.

## 2 RELATED WORKS

**Entropy regularized optimal transport.** A few recent works employ the idea of a coupling with the same marginal constraints. Feydy et al. (2019); Mensch et al. (2019) use static entropy-regularized optimal transportation from a distribution to itself to build a cost function correlated to uncertainty. Sander et al. (2022) reinterpret attention matrices in transformers as transport plans from a distribution to itself, while Agarwal et al. (2024) analyze this reinterpretation in the context of gradient flows. Also relevant is the work of Kurras (2015), who shows that, over discrete state spaces, Sinkhorn's algorithm can be simplified in the case of identical marginal constraints. These works do not consider the coupling with the same marginal constraints from the perspective of path measures on continuous-state spaces. In our paper, we focus on the path measure formulation instead of viewing it as a self-transport map and present a practical algorithm to solve it.

**Expectation maximization.** Our methodology can be broadly categorized under the umbrella of expectation maximization algorithms, drawing from the theory of information geometry. A number of recent papers introduce related formulations to machine learning; most relevant to us are the works of Brekelmans & Neklyudov (2023); Vargas & Nüsken (2023). These works, however, focus on the case of finding a path measure with two distinct marginal constraints, overlooking the potential application to resampling and algorithmic simplifications obtained for the case in which the marginal constraints are the same. In our work, we derive an algorithm that is distinct, yet similar in flavor, to address this overlooked version of the problem, i.e. the mirror Schrödinger bridge.

**Schrödinger bridges and stochastic interpolants.** Schrödinger bridges have been used to obtain generative models by flowing samples from a prior distribution to an empirical data distribution from which new data is to be sampled. Several methods have been proposed to this end: De Bortoli et al. (2021); Vargas et al. (2021) iteratively estimate the drift of the SDE associated with the diffusion processes of half-bridge formulations. While the first uses neural networks and score matching, the latter employs Gaussian processes. From these, a number of extensions or alternative methods have been presented; most relevant are (Shi et al., 2023; Peluchetti, 2023), which extend

(De Bortoli et al., 2021) but differ with respect to the projection sets used to define their half-bridge formulations. Schrödinger bridge based methods alleviate the computational expense incurred by score-based generative models (SGM) (De Bortoli et al., 2021). The latter requires the forward diffusion process to run for longer times with smaller step sizes. Unlike SGM, our method provides a tool to flow an existing sample in the same data distribution with control over the spread of the newly obtained sample.

NEW

To the best of our knowledge, the work of Albergo et al. (2023) is the only one in the literature on generative modeling that maps from a distribution to itself. In their paper, flow matching learns a drift function associated with a stochastic path from the data distribution to itself. Their stochastic interpolants, however, are not optimal with respect to any functional. In particular, they lack optimality in the relative entropy sense, a property correlated to sampling effectiveness and generation quality (Shaul et al., 2023) and hence of practical importance. By contrast, our method discovers the coupling with minimal relative entropy, akin to methods such as (De Bortoli et al., 2021; Shi et al., 2023); our method, however, presents certain algorithmic advantages over these, which can only be derived for the mirror case.

## 3 MATHEMATICAL PRELIMINARIES

**Definition.** Let $n > 0$ be an integer, and let $\mathbb{P}^0 \in \mathcal{P}(C([0,1], \mathbb{R}^n))$ be a reference measure in the space of path measures. Following (Jamison, 1975; Léonard, 2014), we define the *Schrödinger bridge problem* to be the problem of finding a path measure $\mathbb{P}_{\mathrm{SB}}$ interpolating between prescribed initial and final marginals $\pi_0$ and $\pi_1$ that is the closest to the reference measure $\mathbb{P}^0$ with respect to the Kullback-Leibler divergence $D_{\mathrm{KL}}$. To be precise, we define $\mathbb{P}_{\mathrm{SB}}$ to be the solution of the following optimization problem:

$$\mathbb{P}_{\mathrm{SB}} := \underset{\mathbb{P} \in \mathbb{D}(\pi_0, \pi_1)}{\arg\min} \; D_{\mathrm{KL}}\left(\mathbb{P} \parallel \mathbb{P}^0\right), \tag{1}$$

where $\mathbb{D}(\pi_0, \pi_1)$ denotes the set of path measures with marginals $\pi_0$ and $\pi_1$. In other words, we say that $\mathbb{P}_{\mathrm{SB}}$ is the *direct $D_{\mathrm{KL}}$ projection* of $\mathbb{P}^0$ onto the space $\mathbb{D}(\pi_0, \pi_1)$.

The reference path measure $\mathbb{P}^0$ is typically chosen to be associated with a diffusion process, which is defined to be any stochastic process $\mathbf{X}_t$ governed by a forward SDE of the form

$$\mathrm{d}\mathbf{X}_t = f_t(\mathbf{X}_t)\mathrm{d}t + \sigma\mathrm{d}\mathbf{W}_t,$$

where $f_t$ denotes the forward drift function, $\sigma > 0$ is the noise coefficient, and $\mathbf{W}_t$ denotes the Wiener process. Such a process $\mathbf{X}_t$ corresponds to a unique path measure once an initial or final condition is specified. An important aspect of diffusion processes is that their time-reversals are diffusion processes of the same noise coefficient $\sigma$. Specifically, if $\mathbf{X}_t$ is a diffusion process with time-reversal denoted by $\mathbf{Y}_t$, then $\mathbf{Y}_t$ is governed by a backward SDE of the form

$$\mathrm{d}\mathbf{Y}_t = b_t(\mathbf{Y}_t)\mathrm{d}t + \sigma\mathrm{d}\mathbf{W}_t,$$

where $b_t$ denotes the backward drift function (see (Winkler et al., 2023, section 2.3)).

In the case where $\mathbb{P}^0$ arises from a diffusion process, any path measure with finite KL divergence with respect to $\mathbb{P}^0$, including the Schrödinger bridge $\mathbb{P}_{\mathrm{SB}}$, necessarily also arises from a diffusion process with noise $\sigma$ (Vargas et al., 2021). Consequently, by adjusting the initial condition of the reference SDE, we can assume that the reference process $\mathbb{P}^0$ has a prescribed initial marginal $\pi_0$, without changing the solution to (1).

**Iterative Proportional Fitting Procedure.** In the literature, the typical strategy for solving the problem (1) is to apply a general technique known as the *Iterative Proportional Fitting Procedure* (IPFP) (Fortet, 1940; Kullback, 1968). This procedure obtains the Schrödinger bridge by iteratively solving the following pair of half-bridge problems:

$$\mathbb{P}^{2k+1} = \underset{\mathbb{P} \in \mathbb{D}(\cdot, \pi_1)}{\arg\min} D_{\mathrm{KL}}\left(\mathbb{P} \parallel \mathbb{P}^{2k}\right), \; \mathbb{P}^{2k+2} = \underset{\mathbb{P} \in \mathbb{D}(\pi_0, \cdot)}{\arg\min} D_{\mathrm{KL}}\left(\mathbb{P} \parallel \mathbb{P}^{2k+1}\right) \tag{2}$$

where $\mathbb{D}(\cdot, \pi_1)$, respectively, $\mathbb{D}(\pi_0, \cdot)$, denotes the space of path measures with final (resp., initial) marginal fixed to be $\pi_1$ (resp., $\pi_0$). Ruschendorf (1995) proves that the sequence of iterates $\mathbb{P}^k$ converges in total variation to $\mathbb{P}_{\mathrm{SB}}$ as $k \to \infty$. IPFP can be thought of as an extension of Sinkhorn's

algorithm to continuous state spaces, where the rescaling updates characteristic of Sinkhorn are replaced by iterated direct $D_{\mathrm{KL}}$ projections onto sets of distributions with fixed initial or final marginal (Essid & Pavon, 2019).

**Applications.** Suppose $\pi_0$ is given by a data distribution $p_{\mathrm{data}}$ and take $\pi_1$ to be an easy-to-sample distribution $p_{\mathrm{prior}}$, e.g., $p_{\mathrm{prior}} = \mathcal{N}(0, \boldsymbol{I})$. The backward diffusion process associated with $\mathbb{P}_{\mathrm{SB}}$ gives a model for sampling from $p_{\mathrm{data}}$. In practice, the IPFP iterates in (2) can be solved using an algorithm known as the diffusion Schrödinger bridge (DSB), developed by De Bortoli et al. (2021). DSB relies on the following observation, which is a consequence of Girsanov's theorem: $\mathbb{P}^{2k+1}$ is the path measure whose backward drift is equal to the time-reversal of the forward drift of $\mathbb{P}^{2k}$, and $\mathbb{P}^{2k+2}$ is the path measure whose forward drift is equal to the time-reversal of the backward drift of $\mathbb{P}^{2k+1}$. Leveraging this fact, DSB solves for $\mathbb{P}_{\mathrm{SB}}$ by training neural networks to learn the forward and backward drift functions associated with the IPFP iterates.

## 4  MIRROR SCHRÖDINGER BRIDGES

Given a reference path measure $\mathbb{P}^0$ and a prescribed marginal distribution $\pi$, we consider the Schrödinger bridge problem between $\pi$ and itself with respect to $\mathbb{P}^0$. In the case where $\mathbb{P}^0$ is time-symmetric, the Schrödinger bridge will inherit the time-symmetry, in which case we call it the *mirror Schrödinger bridge* from $\pi$ to itself with respect to $\mathbb{P}^0$. Mathematically, we write

$$\mathbb{P}_{\mathrm{MSB}} := \operatorname*{arg\,min}_{\mathbb{P} \in \mathbb{D}(\pi, \pi)} D_{\mathrm{KL}} \left( \mathbb{P} \parallel \mathbb{P}^0 \right), \tag{3}$$

so that $\mathbb{P}_{\mathrm{MSB}} \in \mathbb{D}(\pi, \pi)$ is the path measure with identical prescribed marginals equal to $\pi$ that is closest to the reference measure $\mathbb{P}^0$ with respect to the KL divergence $D_{\mathrm{KL}}$.

A naïve approach to solving the mirror Schrödinger bridge problem (3) is to apply IPFP with both marginals $\pi_0 = \pi_1$ set equal to $\pi$. In practice, this requires iterative training of two neural networks $f_t^\theta$ and $b_t^\phi$, the first modeling the drift of the forward diffusion process associated to $\mathbb{P}_{\mathrm{MSB}}$ and the latter modeling the drift of the corresponding backward process. But this straightforward application of IPFP leads to unnecessary computational expense, as it fails to use the time-symmetry of the problem (3). In particular, at optimality the forward and backward drifts of $\mathbb{P}_{\mathrm{MSB}}$ must be equal, because the mirror Schrödinger bridge $\mathbb{P}_{\mathrm{MSB}}$ is time-symmetric. Related works in entropic optimal transportation suggest that the use of one optimization variable for the static transport formulation in the symmetric case (see (Kurras, 2015, Section 3) and (Feydy et al., 2019, Equations (24)-(25))), but to our knowledge no approach has been developed to leverage symmetry for the dynamical formulation in the language of path measures.

In section 4.1, we develop a method for solving (3) by leveraging time-symmetry in conjunction with a general technique from information geometry known as the *Alternating Minimization Procedure* (AMP), which was first formalized by Csiszár & Tusnády (1984). Then, in section 4.3, we derive an efficient algorithm that involves training a single neural network modeling the drift of the diffusion process associated to $\mathbb{P}_{\mathrm{MSB}}$ and requires half of the computational expense in terms of training iterations for the mirror problem, when compared to other IPFP-based algorithms.

### 4.1  ALTERNATING MINIMIZATION PROCEDURE

Take the reference path measure $\mathbb{P}^0$ to be time-symmetric. As an example, we can take $\mathbb{P}^0$ to be associated to an Ornstein–Uhlenbeck process $\mathbf{X}_t$ given by an SDE of the form $\mathrm{d}\mathbf{X}_t = -\alpha \mathbf{X}_t \mathrm{d}t + \sigma \mathrm{d}\mathbf{W}_t$, for $\alpha > 0$, or more generally any reversible diffusion process. We propose the following iterative scheme:

$$\mathbb{P}^{2k+1} = \operatorname*{arg\,min}_{\mathbb{P} \in \mathbb{D}(\pi, \cdot)} D_{\mathrm{KL}} \left( \mathbb{P} \parallel \mathbb{P}^{2k} \right) \qquad \text{(direct } D_{\mathrm{KL}} \text{ projection)} \tag{4}$$

$$\mathbb{P}^{2k+2} = \operatorname*{arg\,min}_{\mathbb{P} \in \mathbb{S}} D_{\mathrm{KL}} \left( \mathbb{P}^{2k+1} \parallel \mathbb{P} \right), \qquad \text{(reverse } D_{\mathrm{KL}} \text{ projection)} \tag{5}$$

where $\mathbb{S}$ is the set of time-symmetric path measures with no marginal constraints. This scheme is an instance of AMP and differs from IPFP in that it alternates between direct and reverse $D_{\mathrm{KL}}$ projections. To see this, note that (4) is a direct $D_{\mathrm{KL}}$ projection and coincides with the odd-numbered

steps in the IPFP iterations (2), whereas (5) is a *reverse* $D_{\mathrm{KL}}$ projection, as the KL divergence is being computed against the optimization parameter $\mathbb{P}$ instead of the previously produced path measure $\mathbb{P}^{2k+1}$. That is to say, each iteration of the AMP scheme in (4)-(5) is designed to obtain the time-symmetric measure $\mathbb{P}$ that minimizes the objective while remaining close in KL divergence $D_{\mathrm{KL}}$ to the measure obtained in the previous half iteration, which satisfies the initial marginal constraint $\pi$. A theoretical requirement for our proposesd scheme is that the reference measure $\mathbf{P}^0$ be time-symmetric. For this reason, standard Brownian motion cannot be used as a prior.

NEW

It is natural to ask why we consider reverse $D_{\mathrm{KL}}$ projections, as opposed to direct projections, onto the space of symmetric path measures. In fact, replacing (5) with a direct $D_{\mathrm{KL}}$ projection would result in a viable symmetrized variant of IPFP, and by (Ruschendorf, 1995), the resulting iterates would converge in total variation to the mirror Schrödinger bridge. The difficulty is in computing the direct $D_{\mathrm{KL}}$ projection of a path measure onto $\mathbb{S}$. As we will demonstrate in section 4.3, it is considerably easier to compute the reverse $D_{\mathrm{KL}}$ projection onto $\mathbb{S}$, as this particular projection can be done completely analytically.

### 4.2 CONVERGENCE

For the scheme in steps (4)-(5) to be practical, we must prove that the iterates $\mathbb{P}^k$ converge to the mirror Schrödinger bridge $\mathbb{P}_{\mathrm{MSB}}$. The pointwise convergence of schemes like steps (4)-(5) was established by Csiszár & Tusnády (1984) in the special case where the state space is finite. In our setting, however, we work with infinite state spaces of the form $\mathbb{R}^n$ for some dimension $n > 0$. In the following theorem, we prove that the sequence obtained in the AMP scheme converges in total variation to the mirror Schrödinger bridge, without relying on the finiteness assumption for the state space. To our knowledge, this result has not been established previously in the literature.

**Theorem 1.** *Let $\mathbb{P}^k$ be the sequence of path measures obtained via the alternating minimization procedure defined in steps (4)-(5). Then $\mathbb{P}^k$ converges to $\mathbb{P}_{\mathrm{MSB}}$ in total variation as $k \to \infty$. Moreover, the total variation between $\mathbb{P}^k$ and $\mathbb{P}^{k+1}$ decays as $o(1/k)$.*

NEW

Our proof strategy for Theorem 1 is inspired by the convergence proofs for IPFP given in (Ruschendorf, 1995, Proposition 2.1) and (De Bortoli et al., 2021, Theorem 36). The basic idea is to prove that the sequence $\mathbb{P}^k$ is Cauchy with respect to the metric $\delta_{\mathrm{TV}}$ induced by total variation; we then conclude using completeness of the space of path measures together with optimality of the Schrödinger bridge. The crucial distinction between our setting and theirs is that one of our $D_{\mathrm{KL}}$ projections is reversed, which presents an additional complication for establishing convergence to the mirror Schrödinger bridge. To overcome this challenge, we make use of an observation made by Vargas & Nüsken (2023, section 4.1 and proof of Proposition 4.1): in traditional IPFP, we can reverse one or both of the direct $D_{\mathrm{KL}}$ projections (2) while preserving the sequence of iterates obtained. In particular, they prove:

**Lemma 2.** *Let $\pi_0$, $\pi_1$ be probability distributions on $\mathbb{R}^n$, and let $\mathbb{P} \in \mathcal{P}(C([0,1], \mathbb{R}^n))$ be any path measure. Then we have the following identities relating direct to reverse $D_{\mathrm{KL}}$ projections:*

$$\underset{\mathbb{Q} \in \mathbb{D}(\cdot, \pi_1)}{\arg\min} D_{\mathrm{KL}}(\mathbb{Q} \parallel \mathbb{P}) = \underset{\mathbb{Q} \in \mathbb{D}(\cdot, \pi_1)}{\arg\min} D_{\mathrm{KL}}(\mathbb{P} \parallel \mathbb{Q})$$

$$\underset{\mathbb{Q} \in \mathbb{D}(\pi_0, \cdot)}{\arg\min} D_{\mathrm{KL}}(\mathbb{Q} \parallel \mathbb{P}) = \underset{\mathbb{Q} \in \mathbb{D}(\pi_0, \cdot)}{\arg\min} D_{\mathrm{KL}}(\mathbb{P} \parallel \mathbb{Q}).$$

Using Lemma 2, we obtain the following result, which states that the Schrödinger bridge can be equivalently defined in terms of *reverse* $D_{\mathrm{KL}}$. We defer the proof to Appendix A.

**Proposition 3.** *Let $\pi_0$, $\pi_1$ be probability distributions on $\mathbb{R}^n$ and let $\mathbb{Q}^0 \in \mathcal{P}(C([0,1], \mathbb{R}^n))$ be any path measure. Then the Schrödinger bridge $\mathbb{Q}_{\mathrm{SB}}$ with respect to $\mathbb{Q}^0$ is the unique solution to the following pair of optimization problems:*

$$\mathbb{Q}_{\mathrm{SB}} = \underset{\mathbb{Q} \in \mathbb{D}(\pi_0, \pi_1)}{\arg\min} D_{\mathrm{KL}}(\mathbb{Q} \parallel \mathbb{Q}^0) = \underset{\mathbb{Q} \in \mathbb{D}(\pi_0, \pi_1)}{\arg\min} D_{\mathrm{KL}}(\mathbb{Q}^0 \parallel \mathbb{Q}).$$

We are now ready to use Lemma 2 and Proposition 3 to prove Theorem 1.

*Proof of Theorem 1.* Equipped with Lemma 2, we can reverse the $D_{\mathrm{KL}}$ projections in the steps given by (4). Then we obtain the following sequence of pairs of reverse $D_{\mathrm{KL}}$ projections:

$$\mathbb{P}^{2k+1} = \underset{\mathbb{P} \in \mathbb{D}(\cdot, \pi)}{\arg\min} D_{\mathrm{KL}}\left(\mathbb{P}^{2k} \,\|\, \mathbb{P}\right) \qquad \text{(reverse } D_{\mathrm{KL}} \text{ projection)}$$

$$\mathbb{P}^{2k+2} = \underset{\mathbb{P} \in \mathbb{S}}{\arg\min} D_{\mathrm{KL}}\left(\mathbb{P}^{2k+1} \,\|\, \mathbb{P}\right), \qquad \text{(reverse } D_{\mathrm{KL}} \text{ projection)}$$

We apply the Pythagorean theorem for reverse $D_{\mathrm{KL}}$ projections, which shows that

$$D_{\mathrm{KL}}\left(\mathbb{P}^0 \,\|\, \mathbb{P}_{\mathrm{MSB}}\right) = \sum_{i=1}^{\infty} D_{\mathrm{KL}}\left(\mathbb{P}^{i-1} \,\|\, \mathbb{P}^i\right) + \lim_{k \to \infty} D_{\mathrm{KL}}\left(\mathbb{P}^k \,\|\, \mathbb{P}_{\mathrm{MSB}}\right) \qquad (6)$$

Since KL divergences are always nonnegative, the sequence of partial sums in (6) is nondecreasing and bounded, so the sum must converge. Thus, for any $\epsilon > 0$, we can choose $N$ sufficiently large to ensure that $D_{\mathrm{KL}}\left(\mathbb{P}^{n_1} \,\|\, \mathbb{P}^{n_2}\right) \le \epsilon$ for all $n_2 > n_1 > N$. By Pinsker's Inequality, we have that the same property holds with $D_{\mathrm{KL}}$ replaced by $\delta_{\mathrm{TV}}$, i.e., the metric induced by total variation. Thus, the sequence $\mathbb{P}^k$ is Cauchy with respect to $\delta_{\mathrm{TV}}$. Since the space of path measures is complete with respect to this metric, there exists a limit $\mathbb{P}^k \to \mathbb{P}^\star$. But just as we argued in the proof of Proposition 3, we can show that $D_{\mathrm{KL}}\left(\mathbb{P}^0 \,\|\, \mathbb{P}_{\mathrm{MSB}}\right) = D_{\mathrm{KL}}\left(\mathbb{P}^0 \,\|\, \mathbb{P}^\star\right)$, so by uniqueness of the Schrödinger bridge with respect to reverse $D_{\mathrm{KL}}$, as shown in Proposition 3, it follows that $\mathbb{P}_{\mathrm{MSB}} = \mathbb{P}^\star$. Finally, from (6), it follows by applying (De Bortoli et al., 2021, Lemma 38) in conjunction with the results of Csiszár & Tusnády (1984) that $D_{\mathrm{KL}}\left(\mathbb{P}^{i-1} \,\|\, \mathbb{P}^i\right) = o(1/i)$, so Pinsker's inequality implies the claimed rate of convergence. $\qquad\square$

NEW

### 4.3 PRACTICAL ALGORITHM

In this section, we describe an algorithm to solve the mirror Schrödinger bridge problem numerically, based on the AMP scheme that we introduced in section 4.1. We choose our reference path measure $\mathbb{P}^0 \in \mathbb{S}$ to be associated to an Ornstein-Uhlenbeck process $\mathbf{X}_t$ given by an SDE of the form $\mathrm{d}\mathbf{X}_t = -\alpha \mathbf{X}_t \mathrm{d}t + \sigma \mathrm{d}\mathbf{W}_t$, for $\alpha > 0$.

Recall that our proposed AMP scheme alternates between direct $D_{\mathrm{KL}}$ projections on the set of path measures with a prescribed initial marginal distribution $\pi$ and reverse $D_{\mathrm{KL}}$ projections on the set of time-symmetric path measures. We now explain how each of these projections is computed in practice. Our algorithm then follows by iteratively applying this pair of projections and is summarized in Algorithm 1.

---

**Algorithm 1** MIRROR SCHRÖDINGER BRIDGE

1: **for** $k \in \{0, \dots, K-1\}$ **do**
2:      **while** not converged **do**
3:          Sample $\mathbf{X}_0^j \sim \pi$ and $\sigma^j \in \mathbb{R}$ from $[\sigma_{\min}, \sigma_{\max}]$ for $j \in \{0, \dots, M-1\}$.
4:          Compute trajectories $\{\mathbf{X}_i^j\}_{i,j=0}^{M-1,N-1}$ via (10) using $f(x) = v_t^{\theta^{2k}}(x)$ as in (12).
5:          Do gradient step on $\theta^{2k+1}$ using (11).
6:      **end while**
7: **end for**
8: **Output:** $v_t^{\theta^\star}$

---

**Direct $D_{\mathrm{KL}}$ projection.** We can compute the $D_{\mathrm{KL}}$ projection onto the set of path measures with a prescribed initial marginal distribution $\pi$ following the trajectory-caching method developed and applied in (Vargas et al., 2021; De Bortoli et al., 2021). Let $\pi$ be a probability distribution on $\mathbb{R}^n$, and let $\mathbb{P} \in \mathbb{D}(\pi, \cdot)$ and $\mathbb{P}^\dagger \in \mathbb{S}$ be path measures corresponding to diffusion processes. Write $f_t^{\mathbb{P}}$ and $b_t^{\mathbb{P}}$ for the forward and backward drift functions corresponding to $\mathbb{P}$, and write $v_t^{\mathbb{P}^\dagger} = f_t^{\mathbb{P}^\dagger} = b_t^{\mathbb{P}^\dagger}$ for the drift of $\mathbb{P}^\dagger$. As a consequence of Girsanov's theorem, we can write $D_{\mathrm{KL}}(\mathbb{P} \,\|\, \mathbb{P}^\dagger)$ explicitly in terms of $f_t^{\mathbb{P}}$ and $v_t^{\mathbb{P}^\dagger}$, or equivalently in terms of $b_t^{\mathbb{P}}$ and $v_t^{\mathbb{P}^\dagger}$; for references, see (Chen et al., 2016, section 3) as well as (Winkler et al., 2023, sections 2.2, 2.3). Indeed, for some constants $C_1$, $C_2$, we have

$$D_{\mathrm{KL}}(\mathbb{P} \,\|\, \mathbb{P}^\dagger) = C_1 + \frac{1}{2\sigma^2} \int_0^1 \mathbb{E}_{\mathbb{P}}\left[\left(f_t^{\mathbb{P}}(\mathbf{X}_t) - v_t^{\mathbb{P}^\dagger}(\mathbf{X}_t)\right)\right]^2 \mathrm{d}t \qquad (7)$$

$$= C_2 + \frac{1}{2\sigma^2} \int_0^1 \mathbb{E}_{\mathbb{P}}\left[\left(b_t^{\mathbb{P}}(\mathbf{X}_t) - v_t^{\mathbb{P}^\dagger}(\mathbf{X}_t)\right)\right]^2 \mathrm{d}t. \qquad (8)$$

In light of the identities (7) and (8), and because drift functions are much more amenable to modeling and estimation than path measures, it is convenient to recast the steps of our AMP scheme as iterative computations of drift functions associated to $D_{\mathrm{KL}}$ projections.

It follows immediately from (7) that the direct $D_{\text{KL}}$ projection of $\mathbb{P}^\dagger$ onto the space $\mathbb{D}(\pi, \cdot)$ is given by the unique path measure $\mathbb{P}$ with initial marginal $\pi$ and forward drift $f_t^{\mathbb{P}}$ equal to the drift $v_t^{\mathbb{P}^\dagger}$ of $\mathbb{P}^\dagger$. In our AMP scheme, we employ this by taking $\mathbb{P} = \mathbb{P}^{2k+1} \in \mathbb{D}(\pi, \cdot)$ to have drift equal to that of $\mathbb{P}^\dagger = \mathbb{P}^{2k}$ for each $k \geq 0$. But as we will see in our analysis of the reverse $D_{\text{KL}}$ projection, it does not suffice for us to know only the forward drift associated to our path measure iterates. We need to know the backward drift $b_t^{\mathbb{P}}$ too, but in practice, we do not have access to it. We use trajectory caching to estimate the backward drift $b_t^{\mathbb{P}}$. Trajectory caching is principled on the fact that $b_t^{\mathbb{P}}$ can be expressed in terms of the expected rate of change in $\mathbf{X}_t$ over time. Concretely, we have the following formula, which can be taken as a formal definition of the backward drift of a diffusion process:

$$b_{1-t}^{\mathbb{P}}(x) = \lim_{\gamma \to 0} \mathbb{E}\left[ \frac{\mathbf{X}_{t-\gamma} - \mathbf{X}_t}{\gamma} \,\middle|\, \mathbf{X}_t = x \right]. \tag{9}$$

To apply (9) in practice, take a positive integer $M$ and let $\{\gamma_i\}_{i=1}^M$ be a sequence of $M$ discrete time steps with sequence of partial sums $\{\bar{\gamma}_i\}_{i=1}^M$. Then we construct a discrete representation of the stochastic process $\mathbf{X}_t$ by using the Euler-Maruyama method to generate a collection of $N$ sample trajectories $\{\mathbf{X}_i^j\}_{i,j=0}^{M-1,N-1}$ starting at the initial distribution $\pi$ in accordance with the SDE $d\mathbf{X}_t = f_t^{\mathbb{P}}(\mathbf{X}_t)dt + \sigma d\mathbf{W}_t$, where we know the forward drift $f_t^{\mathbb{P}}$ because we matched it to the drift of $\mathbb{P}^\dagger$. Explicitly, we have for all $i \in \{0, \ldots, M-2\}$ and $j \in \{0, \ldots, N-1\}$ that

$$\mathbf{X}_{i+1}^j = \mathbf{X}_i^j + f_{\bar{\gamma}_i}^{\mathbb{P}}(\mathbf{X}_i^j)\gamma_i + \sigma^j\sqrt{\gamma_i}\mathbf{Z}_i^j, \quad \text{where} \quad \mathbf{Z}_i^j \sim \mathcal{N}(0, \mathbf{I}). \tag{10}$$

The limiting quantity in (9) is then leveraged as the target of the loss function used to train a neural network $v_t^\theta$, which approximates the backward drift $b_t^{\mathbb{P}}$ for a specified range of $\sigma$ values $[\sigma_{\min}, \sigma_{\max}]$. Specifically, we define the following loss function in terms of the optimization parameter $\theta$:

$$\ell(\theta) = \frac{1}{N} \sum_{i=1}^{M-1} \sum_{j=0}^{N-1} \left\| v_{\bar{\gamma}_{i+1}}^\theta(\mathbf{X}_{i+1}^j) - \frac{\mathbf{X}_i^j - \mathbf{X}_{i+1}^j}{\gamma_{i+1}} - \left( f_{\bar{\gamma}_i}^{\mathbb{P}^\dagger}(\mathbf{X}_{i+1}^j) - f_{\bar{\gamma}_i}^{\mathbb{P}^\dagger}(\mathbf{X}_i^j) \right) \right\|^2 \tag{11}$$

Observe that the first two terms in the loss constitute the difference between the drift and the infinitesimal rate of change of the process $\mathbf{X}_t$, i.e., the discretization of the difference between the left- and right-hand sides of (9). The network parameters $\theta$ are then learned via gradient descent with respect to the loss function $\ell(\theta)$. The resulting function $v_t^\theta$, where $\theta$ minimizes the loss $\ell(\theta)$, approximates the desired backward drift, as is suggested by De Bortoli et al. (2021, Proposition 3).

**Reverse $D_{\text{KL}}$ projection.** We now describe how to compute the reverse $D_{\text{KL}}$ projection onto the set $\mathbb{S}$ of time-symmetric path measures. We are interested in computing the associated time-symmetric drift, rather than the path measure itself. To this end, let $\pi$ be a probability distribution on $\mathbb{R}^n$, and let $\mathbb{P} \in \mathbb{D}(\pi, \cdot)$ and $\mathbb{P}^\dagger \in \mathbb{S}$ be path measures corresponding to diffusion processes. Suppose we seek to minimize $D_{\text{KL}}(\mathbb{P} \parallel \mathbb{P}^\dagger)$ over all $\mathbb{P}^\dagger \in \mathbb{S}$. Using (7) and (8), we can write $D_{\text{KL}}(\mathbb{P} \parallel \mathbb{P}^\dagger)$ explicitly in terms of the forward and backward drift functions of the SDE corresponding to the path measures $\mathbb{P}$ and $\mathbb{P}^\dagger$. A key benefit of considering the reverse $D_{\text{KL}}$ projection is that the expectation values in (7) and (8) are taken with respect to the fixed path measure $\mathbb{P}$, and not with respect to the varying path measure $\mathbb{P}^\dagger$. This allows us to apply calculus of variations to compute a closed-form expression for the drift of the minimizer of $D_{\text{KL}}(\mathbb{P} \parallel \mathbb{P}^\dagger)$ over $\mathbb{P}^\dagger \in \mathbb{S}$. First, observe that we can combine (7) and (8) to rewrite $D_{\text{KL}}(\mathbb{P} \parallel \mathbb{P}^\dagger)$ in a time-symmetric formulation as follows:

$$D_{\text{KL}}(\mathbb{P} \parallel \mathbb{P}^\dagger) = C + \frac{1}{4\sigma^2} \int_0^1 \mathbb{E}_{\mathbb{P}}\left[ \left( f_t^{\mathbb{P}}(\mathbf{X}_t) - v_t^{\mathbb{P}^\dagger}(\mathbf{X}_t) \right)^2 + \left( b_t^{\mathbb{P}}(\mathbf{X}_t) - v_t^{\mathbb{P}^\dagger}(\mathbf{X}_t) \right)^2 \right] dt,$$

where $C$ is a constant. Note that the sum of squares inside the expectation on the right-hand side above is always nonnegative. Consequently, to minimize $D_{\text{KL}}(\mathbb{P} \parallel \mathbb{P}^\dagger)$, it suffices to choose $v_t^{\mathbb{P}^\dagger}$ so that it minimizes this sum of squares pointwise everywhere. Taking the first variation of this sum of squares with respect to $v_t^{\mathbb{P}^\dagger}$, setting the resulting expression equal to zero, and solving for the optimal $v_t^{\mathbb{P}^\dagger}$, we find that

$$v_t^{\mathbb{P}^\dagger}(x) = \frac{1}{2}\left( f_t^{\mathbb{P}}(x) + b_t^{\mathbb{P}}(x) \right). \tag{12}$$

That is, the choice of $\mathbb{P}^\dagger \in \mathbb{S}$ minimizing $D_{\text{KL}}(\mathbb{P} \parallel \mathbb{P}^\dagger)$ has drift function given by the average of the forward and backward drifts of $\mathbb{P}$.

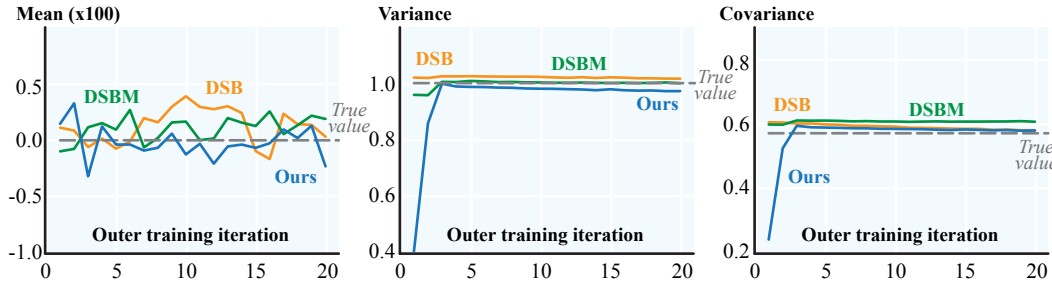

Figure 1: For each method, we plot the mean (left) and variance (middle) obtained for the terminal samples, i.e. samples obtained at time $t = T$, as well as the covariance (right) of the joint distribution, versus the number of outer iterations, averaged over 5 trials.

In our AMP scheme, we employ (12) by taking the forward and backward drifts corresponding to $\mathbb{P} = \mathbb{P}^{2k+1}$ and averaging them to obtain the drift of the symmetric path measure $\mathbb{P}^{\dagger} = \mathbb{P}^{2k+2}$. Practically speaking, if the drift of $\mathbb{P}^{2k}$ is a parametrized by a neural network $v_t^{\theta^k}$ for each $k$, we take the drift of $\mathbb{P}^{2k+1}$ to be the average of the outputs of the neural networks $v_t^{\theta^{k-1}}$ and $v_t^{\theta^k}$. In Algorithm 1, we denote the limiting drift as $v_t^{\theta^\star}$.

### 4.4 Sampling with In-distribution Variation

In this section, we provide a short intuitive explanation of how our method allows for resampling with prescribed proximity to an input sample. Given such a sample $x_0 \sim \pi$, we solve the SDE corresponding to the Schrödinger bridge to push $x_0$ forward in time, arriving at a final sample $x_1 \in \pi$. We want $x_1$ to be a variation of $x_0$, where the proximity of $x_1$ to $x_0$ correlates with the size of the noise coefficient $\sigma$. Justifying this mathematically requires understanding how the conditional distribution $\mathbf{X}_1 \mid \mathbf{X}_0 = x_0$, specifically its mean and variance, depend on $\sigma$. While these quantities do not in general have closed form expressions, it is possible to compute them exactly in the case where $\pi = \mathcal{N}(0, \mathbf{I})$ is a 1-dimensional Gaussian.

In this case, let $\mathbf{X}_t$ denote the diffusion process associated to the Schrödinger bridge, where the reference path measure corresponds to an Ornstein-Uhlenbeck reference process with drift coefficient $-\alpha$. In Proposition 4 (see Appendix B for the statement and proof) we determine the joint distribution of $\mathbf{X}_0$ and $\mathbf{X}_1$ in terms of a quantity $\beta$, which is a function of $\alpha$ and $\sigma$ that grows approximately as $1 + c(\alpha) \times \sigma^2$ for some function $c$. Let $p(x, y)$ denote the probability density function of the joint distribution of $\mathbf{X}_0$ and $\mathbf{X}_1$, and recall that $p(x, y)$ is the product of the conditional PDF of $\mathbf{X}_1 \mid \mathbf{X}_0$ with the PDF of $\mathbf{X}_0$. Using this fact in conjunction with Proposition 4, the PDF of $\mathbf{X}_1 \mid \mathbf{X}_0 = x_0$ is

$$ p_{\mathbf{X}_1 \mid \mathbf{X}_0 = x_0}(y) = p(x_0, y) / p_{\mathbf{X}_0}(x_0) = e^{-\frac{1}{2(1 - \beta^2)}(x_0^2 - 2\beta x_0 y + y^2) + \frac{x_0^2}{2}}. $$

From the right-hand side, we see that $\mathbf{X}_1 \mid \mathbf{X}_0 = x_0$ is Gaussian with mean and variance given by

$$ \mathbb{E}[\mathbf{X}_1 \mid \mathbf{X}_0 = x_0] = x_0 \left( \beta / 1 - \beta^2 \right), \ \mathbb{E}\left[ (\mathbf{X}_1 - \mathbb{E}[\mathbf{X}_1 \mid \mathbf{X}_0 = x_0])^2 \mid \mathbf{X}_0 = x_0 \right] = 1 - \beta^2. $$

Thus, changing the noise value $\sigma$ alters both the mean and variance of samples pushed forward via the Schrödinger bridge. Indeed, in the case of the mean, it grows inversely proportional to $\sigma^2$. Consequently, if $\sigma < 1$, then we should expect the Schrödinger bridge to push samples away from the distribution mean, whereas if $\sigma > 1$, then the opposite occurs, and samples experience mean reversion. As for the variance, note that $1 - \beta^2$ grows at least as fast as $\sigma^2$, so we should expect the Schrödinger bridge to produce samples with spread that increases as $\sigma$ increases. We expect that similar effects occur even when the marginal distribution $\pi$ is not Gaussian: i.e., the value of $\sigma$ should be directly related to the proximity of generated samples in an analogous way.

## 5 Experiments

We demonstrate the flexibility of our method on a number of conditional sampling tasks. We first show numerical convergence against the solution of the mirror Schrödinger bridge in a case where

an analytical solution is available. Next, we consider resampling from 2-dimensional datasets and demonstrate control over the in-distribution variation of new data points, which is an added feature of our method. Lastly, we provide examples of image resampling, illustrating how our method can be used to produce image variations with control over the proximity to the original.

**Gaussian Transport.** We start by comparing our method with two alternative algorithms, DSB (De Bortoli et al., 2021) and DSBM (Shi et al., 2023), when applied to the mirror Schrödinger bridge case on Gaussians of varying dimension. Figure 1 shows that, in the case of dimension $d = 50$, as the number of outer iterations increases, the empirical convergence of our method performs on par with both DSB and DSBM with the added benefit that each outer iteration with our algorithm requires half the training iterations. Recall that our method trains a single neural network to model a time-symmetrized drift function $v_t^\theta$ rather than a neural network for each of the forward and backward drift functions. More details on the derivation of the analytical solution for this experiment, as well as information on parameters, can be found in Appendix B. Additional results for dimensions $d = 5, 20$ can be found in Figure 6.

**2D Datasets.** To illustrate the behavior of our method, we use our algorithm to re-sample from 2-dimensional distributions. Unlike the mirror Schrödinger bridge with Gaussians, an analytical solution for mirror bridge with these more general distributions is not known. We consider learning the drift function $v_t^\theta$ associated with the mirror Schrödinger bridge that flows samples from $p_{\text{data}}$ to itself. The goal is to obtain new samples that are in the distribution $p_{\text{data}}$ but exhibit some level of variation, i.e., in-distribution variation, correlated to the noise coefficient $\sigma$ in the diffusion process. Note that computing mirror Schrödinger bridges with a range of noise values by training one neural network is not possible using existing alternative methods.

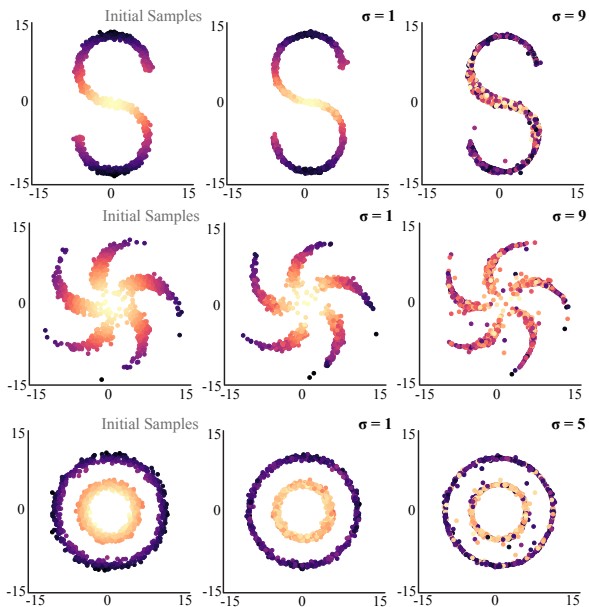

Figure 2: Samples obtained using our method with varying values of $\sigma$. Samples are colored based on initial position.

In the columns of Figure 2, we show the result of flowing samples via the mirror Schrödinger bridge with varying values of noise. We observe that the in-distribution variation of data points is controlled by the choice of $\sigma$ value, which can indeed be detected by the mixing of colors, or lack of thereof, in each terminal distribution shown. For instance, in the bottom row, we find mixing from samples between the inner and outer circles with the largest value of $\sigma$, compared with no mixing of samples between circles with the smallest value of sigma.

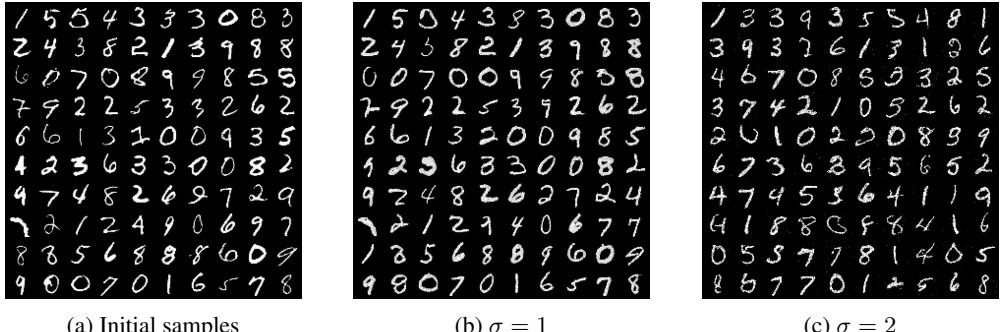

| (a) Initial samples | (b) $\sigma = 1$ | (c) $\sigma = 2$ |
|---|---|---|

Figure 3: Samples produced by the mirror Schrödinger bridges for the empirical distribution of handwritten digits, using varying levels of noise.

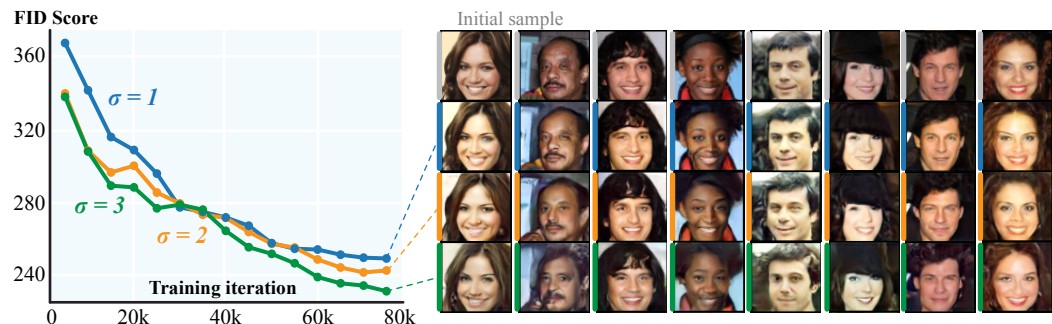

Figure 4: The control over in-distribution variance effect of $\sigma$ for a variety of initial samples (first row) from the empirical distribution of images in CelebA.

**Image Resampling.** We also train our algorithm on each of the MNIST, CelebA, and Flower102 datasets. Details on training parameters and architecture for all experiments using images can be found in Appendix C. Our results show that mirror Schrödinger bridges can be used to produce new samples from an image dataset with control over the proximity to the initial sample. In Figure 3, we resample from MNIST using varying levels of noise. We find that pushforward images obtained with a lower fixed value of noise (3b) are visually closer to the initial images (3a) obtained with a higher fixed value of noise (3c).

Figure 4 demonstrates the same control over the in-distribution variation of pushforward samples using the RGB dataset CelebA. In each column, we exhibit a different sample from the dataset and, in each row, we show the corresponding pushforward obtained for different noise values. These results can be obtained without retraining the neural network. The typical metric to assess resampling quality for the image generation case is the Fréchet inception distance (FID) score, which we have plotted against training iterations. We observe FID scores decreasing with training iterations.

FIX

NEW

Figure 12 includes more results using the CelebA dataset, and Figure 7 shows the nearest neighbors in the dataset to the generated images. In the latter Figure, as desired, the nearest neighbor of the generated sample is the initial sample itself, and the generated sample is distinct from all of its nearest neighbors, showing that our model does not simply regurgitate nearest neighbors of the initial sample as proximal outputs.

Figure 5 highlights how mirror Schrödinger bridges can be used as a flexible and well-principled tool to perform small edits to RGB images while guaranteeing the result to be in-distribution. This task can be performed by choosing an appropriately small value for $\sigma$.

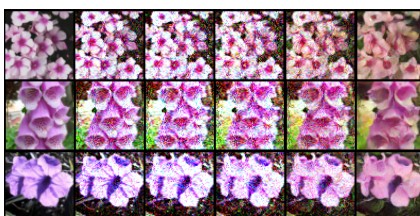

Figure 5: The effect of using small $\sigma$ when pushing samples forward. From left to right: initial samples, intermediate times, and samples at terminal time.

# 6 CONCLUSION

By studying an overlooked version of the Schrödinger bridge problem, which we coin the mirror Schrödinger bridge, we present an algorithm to sample with control over the in-distribution variation of new data points. Our method is flexible and requires fewer training iterations than existing alternatives (De Bortoli et al., 2021; Shi et al., 2023) designed for the general Schrödinger bridge problem. From a theoretical perspective, our method presents advantages over mirror interpolants (Albergo et al., 2023), specifically by obtaining kinetic optimality. While one might consider optimizing fixed mirror interpolants, the resulting min-max optimization problem is intractable (Shaul et al., 2023). By contrast, our method is numerically tractable, is well-principled, and cuts down training in applications where control over in-distribution variation is desired. On the application front, we demonstrate that our method is a flexible tool to obtain new data points from empirical distributions in a variety of domains, including 2-dimensional measures and image datasets. In future work, we hope to study of a potential $\sigma$ threshold for a sample to change class when resampled or, in the same direction, to make class a neural network input, similar to text prompting in image generation.

NEW

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

## A PROOF OF PROPOSITION 3

The first claimed expression is the very definition of $\mathbb{Q}_{\mathrm{SB}}$. As for the second claimed expression, let $\mathbb{Q}^k$ be the sequence of IPFP iterates. Note that by Lemma 2, we have

$$\mathbb{Q}^{2k+1} = \underset{\mathbb{P} \in \mathbb{D}(\cdot, \pi_1)}{\arg\min} D_{\mathrm{KL}}\left(\mathbb{Q}^{2k} \parallel \mathbb{Q}\right), \; \mathbb{Q}^{2k+2} = \underset{\mathbb{P} \in \mathbb{D}(\pi_0, \cdot)}{\arg\min} D_{\mathrm{KL}}\left(\mathbb{Q}^{2k+1} \parallel \mathbb{Q}\right).$$

Since $\mathbb{Q}_{\mathrm{SB}}$ belongs to both projection sets, the Pythagorean theorem for reverse $D_{\mathrm{KL}}$ projections (Brekelmans & Neklyudov, 2023, Theorem 3.4) (see also (Csiszár & Matus, 2003, Theorem 5)) yields that for each $k$ we have

$$D_{\mathrm{KL}}\left(\mathbb{Q}^0 \parallel \mathbb{Q}_{\mathrm{SB}}\right) = \sum_{i=1}^{k} D_{\mathrm{KL}}\left(\mathbb{Q}^{i-1} \parallel \mathbb{Q}^i\right) + D_{\mathrm{KL}}\left(\mathbb{Q}^k \parallel \mathbb{Q}_{\mathrm{SB}}\right) \tag{13}$$

Now the sequence $D_{\text{KL}}(\mathbb{Q}^k \parallel \mathbb{Q}_{\text{SB}})$ converges to zero because the sequence $D_{\text{KL}}(\mathbb{Q}_{\text{SB}} \parallel \mathbb{Q}^k)$ converges to zero (see, e.g., (Weis, 2014, Theorem 3.21.4)). Thus, taking the limit as $k \to \infty$, we deduce that

$$D_{\text{KL}}\left(\mathbb{Q}^0 \parallel \mathbb{Q}_{\text{SB}}\right) = \sum_{i=1}^{\infty} D_{\text{KL}}\left(\mathbb{Q}^{i-1} \parallel \mathbb{Q}^i\right)$$

Now, write $\mathbb{Q}^{\star} = \arg\min_{\mathbb{Q} \in \mathbb{D}(\pi_0, \pi_1)} D_{\text{KL}}\left(\mathbb{Q}^0 \parallel \mathbb{Q}\right)$. A similar argument shows that

$$D_{\text{KL}}\left(\mathbb{Q}^0 \parallel \mathbb{Q}^{\star}\right) = \sum_{i=1}^{\infty} D_{\text{KL}}\left(\mathbb{Q}^{i-1} \parallel \mathbb{Q}^i\right) + \lim_{k \to \infty} D_{\text{KL}}\left(\mathbb{Q}^k \parallel \mathbb{Q}^{\star}\right)$$
$$= D_{\text{KL}}\left(\mathbb{Q}^0 \parallel \mathbb{Q}_{\text{SB}}\right) + \lim_{k \to \infty} D_{\text{KL}}\left(\mathbb{Q}^k \parallel \mathbb{Q}^{\star}\right) \geq D_{\text{KL}}\left(\mathbb{Q}^0 \parallel \mathbb{Q}_{\text{SB}}\right),$$

where the last inequality above follows from the nonnegativity of the KL divergence. It follows that $D_{\text{KL}}\left(\mathbb{Q}^0 \parallel \mathbb{Q}_{\text{SB}}\right)$ also achieves the desired minimum $D_{\text{KL}}$, i.e., we have $D_{\text{KL}}\left(\mathbb{Q}^0 \parallel \mathbb{Q}_{\text{SB}}\right) = D_{\text{KL}}\left(\mathbb{Q}^0 \parallel \mathbb{Q}^{\star}\right)$. Finally, we must rule out the possibility that this minimizer is not unique. To do this, observe that, by the squeeze theorem, we must have

$$\lim_{k \to \infty} D_{\text{KL}}\left(\mathbb{Q}^k \parallel \mathbb{Q}^{\star}\right) = 0.$$

We can now apply Pinsker's Inequality, which tells us that the KL divergence $D_{\text{KL}}$ is at least a constant multiple of the square of the metric $\delta_{\text{TV}}$ induced by total variation. More precisely, we have that $D_{\text{KL}}\left(\mathbb{Q}^k \parallel \mathbb{Q}^{\star}\right) \geq 2\delta_{\text{TV}}\left(\mathbb{Q}^k, \mathbb{Q}^{\star}\right)^2$. We deduce that

$$\lim_{k \to \infty} \delta_{\text{TV}}\left(\mathbb{Q}^k, \mathbb{Q}^{\star}\right) = 0,$$

which implies that $\mathbb{Q}^k$ converges to $\mathbb{Q}^{\star}$ in total variation. We conclude that $\mathbb{Q}^{\star} = \mathbb{Q}_{\text{SB}}$. $\qquad\square$

## B  ANALYTICAL SOLUTION FOR GAUSSIAN EXPERIMENT

**Proposition 4.** *Consider the static Schrödinger bridge problem with initial and final marginals equal to the $d$-dimensional Gaussian distribution with zero mean and unit variance, where we take the reference measure $\pi^0$ corresponding to the OU process $d\mathbf{X}_t = -\alpha\mathbf{X}_t dt + \sigma d\mathbf{W}_t$ running from $t = 0$ to $t = 1$. The solution $\pi^*$ to this problem is a $2d$-dimensional Gaussian with zero mean and covariance matrix $\Sigma$ given by*

$$\Sigma = \begin{pmatrix} \Sigma_{00} & \Sigma_{01} \\ \Sigma_{10} & \Sigma_{11} \end{pmatrix} = \begin{pmatrix} \boldsymbol{I} & \beta\boldsymbol{I} \\ \beta\boldsymbol{I} & \boldsymbol{I} \end{pmatrix}, \quad where \quad \beta = \frac{\sigma^2(1 - e^{2\alpha}) + \sqrt{16e^{2\alpha}\alpha^2 + \sigma^4\left(1 - e^{2\alpha}\right)^2}}{4\alpha e^{\alpha}}$$

*Proof.* We follow the proof of (De Bortoli et al., 2021, Proposition 46), which established the corresponding result in the case where the reference process has zero drift. Imitating the proof of (De Bortoli et al., 2021, Proposition 43), we see that the static Schrödinger bridge $\pi^*$ exists and is a $2d$-dimensional Gaussian. That the mean equals zero follows from the fact that both marginals have zero mean. The rest of the proof is devoted to determining the covariance matrix $\Sigma$ of $\pi^*$.

The fact that marginals have unit variance implies that $\Sigma_{00} = \Sigma_{11} = \boldsymbol{I}$. To compute $\Sigma_{01}$ and $\Sigma_{10}$, we start by computing the probability density function (PDF) $p^0(x, y)$ of the reference measure $\pi^0$, where $x, y \in \mathbb{R}^d$. Recall that $p^0(x, y)$ is the product of the conditional PDF of $\mathbf{X}_1 \mid \mathbf{X}_0$ with the PDF of $\mathbf{X}_0$. Thus, we have

$$p^0(x, y) = p_{\mathbf{X}_1 \mid \mathbf{X}_0}(x, y) \times p_{\mathbf{X}_0}(x).$$

Note that $\mathbf{X}_0$ has zero mean and unit variance, so up to normalization we have

$$p_{\mathbf{X}_0}(x) \propto e^{-\frac{x^2}{2}}.$$

On the other hand, the mean and variance of the conditional distribution $\mathbf{X}_1 \mid \mathbf{X}_0$ are computed in (Trajanovski et al., 2023, section II), where it is shown that they are respectively given by

$$xe^{-\alpha} \quad \text{and} \quad \sigma_1^2 := \frac{\sigma^2}{2\alpha}(1 - e^{-2\alpha}).$$

It follows that
$$p_{\mathbf{X}_1|\mathbf{X}_0}(x,y) \propto e^{-\frac{1}{2\sigma_1^2}\left(y - e^{-\alpha}x\right)^2}.$$

Combining these calculations, we conclude that the joint distribution has PDF given by
$$p^0(x,y) \propto e^{-\frac{1}{2}\left((1+\sigma_1^{-2}e^{-2\alpha})x^2 - 2\sigma_1^{-2}e^{-\alpha}xy + \sigma_1^{-2}y^2\right)}.$$

This distribution is evidently a Gaussian with zero mean and covariance matrix $\Sigma^0$ given by
$$\Sigma^0 = \begin{pmatrix} \boldsymbol{I} & e^{-\alpha}\boldsymbol{I} \\ e^{-\alpha}\boldsymbol{I} & (\sigma_1^2 + e^{-2\alpha})\boldsymbol{I} \end{pmatrix}.$$

Note in particular that the variance of the marginal of $\pi^0$ at $t = 1$ is equal to the coefficient of the bottom-right entry of $\Sigma^0$, which is $\sigma_1^2 + e^{-2\alpha}$. Now, the KL divergence between a 2-dimensional Gaussian distribution $\widetilde{\pi}$ with zero mean and covariance matrix $\widetilde{\Sigma}$ and the distribution $\pi^0$ is given explicitly by
$$D_{\mathrm{KL}}(\widetilde{\pi} \parallel \pi^0) = \frac{1}{2}\left(\log \frac{\det \Sigma^0}{\det \widetilde{\Sigma}} - d + \mathrm{Tr}\left(\Sigma^{0^{-1}}\widetilde{\Sigma}\right)\right).$$

If we take $\widetilde{\Sigma}$ to be of the form
$$\widetilde{\Sigma} = \begin{pmatrix} \boldsymbol{I} & S \\ S^T & \boldsymbol{I} \end{pmatrix},$$

which matches the form of the covariance $\Sigma$ for $\pi^*$, then
$$D_{\mathrm{KL}}(\widetilde{\pi} \parallel \pi^0) = \frac{1}{2}\left(-\log \det \widetilde{\Sigma} - 2e^{-\alpha}\sigma_1^{-2}\,\mathrm{Tr}(S) + C\right)$$

where $C \in \mathbb{R}$ is a nonzero constant independent of $\widetilde{\Sigma}$. As argued in (De Bortoli et al., 2021, proof of Proposition 46), we can assume $S = S^T$ is a symmetric matrix, as doing so will only decrease $D_{\mathrm{KL}}(\widetilde{\pi} \parallel \pi^0)$, so $S$ is diagonalizable. Let $\lambda_1, \ldots, \lambda_d$ denote the eigenvalues of $S$, counted with multiplicity. Using the well-known formula for the determinant of a block $2 \times 2$ matrix, we find that
$$\det \widetilde{\Sigma} = \det(\boldsymbol{I} - S^2) = \prod_{i=1}^d (1 - \lambda_i^2).$$

Thus, we obtain
$$D_{\mathrm{KL}}(\widetilde{\pi} \parallel \pi^0) = \frac{1}{2}\sum_{i=1}^d f(\lambda_i) + C, \quad \text{where} \quad f(x) = -\log(1 - x^2) - 2e^{-\alpha}\sigma_1^{-2}x.$$

Note in particular that since $\widetilde{\Sigma}$ is a covariance matrix, it is positive semi-definite, and so its eigenvalues $1 - \lambda_i^2$ must be nonnegative, implying that $|\lambda_i| \leq 1$ for each $i$.

Minimizing $D_{\mathrm{KL}}(\widetilde{\pi} \parallel \pi^0)$ then amounts to take $\lambda_1 = \cdots = \lambda_d = \beta$ in such a way that $f(\beta)$ is minimized. Observe that the equation
$$f'(\beta) = \frac{2\beta}{1 - \beta^2} - 2e^{-\alpha}\sigma_1^{-2} = 0$$

is solved by
$$\beta = \frac{\sigma^2(1 - e^{2\alpha}) \pm \sqrt{16e^{2\alpha}\alpha^2 + \sigma^4\left(1 - e^{2\alpha}\right)^2}}{4\alpha e^\alpha}.$$

We then choose the sign to be $+$ to ensure that $|\beta| \leq 1$. $\qquad\square$

## C  IMPLEMENTATION DETAILS

In this section we give further details on our experimental setup. Akin to Song & Ermon (2020, Technique 5) and De Bortoli et al. (2021, Technique 6), we improve performance of Algorithm 1 by implementing the exponential moving average (EMA) of network parameters.

NEW

### C.1 GAUSSIAN TRANSPORT

We use the MLP large network from (De Bortoli et al., 2021) for DSB and DSBM in all Gaussian transport experiments. For our method, we modify this network to take $\sigma$ as an input. The values of $\sigma$ are uniformly sampled from the (inclusive) interval from 1 to 5 for training, and at test time we fix $\sigma = 1$ for all samples to compare with DSB and DSBM, which do not take $\sigma$ as a network input, but each use $\sigma = 1$ via the SDE discretization. We run the same experiment for dimension $d = 5$ and $d = 20$ (in Figure 6), and $d = 50$ (in Figure 1). The number of samples for all experiments is 10,000. We use 20 timesteps and train for 10,000 inner iterations for each of 20 outer iterations.

### C.2 2D DATASETS

We modify the network architecture with positional encoding from (Vaswani et al., 2017), which is used by De Bortoli et al. (2021), to take values of noise $\sigma$ rather than tuples of only $\mathbf{X}$ and $t$. The values of $\sigma$ are concatenated to the spatial features before the first MLP block is applied. This modified network is used to parametrize our drift function. We use Adam optimizer with learning rate $10^{-4}$ and momentum 0.9. We train each example for 10,000 inner iterations per outer iteration of the algorithm. Figure 2 shows the terminal samples obtained for outer iteration 30 for all example datasets. The noise values $\sigma^j$ are sampled uniformly in the range from 1 to 9 for training. At test time, a fixed $\sigma$ value is chosen for all sample trajectories. We train with 10,000 samples, which are refreshed each 1,000 iterations. We use 20 timesteps of size 0.01 each. All 2-dimensional experiments run on CPU.

### C.3 IMAGE RESAMPLING

For the image dataset experiments, we modify the U-Net architecture used in (De Bortoli et al., 2021; Shi et al., 2023) to take values of noise $\sigma$. Each value $\sigma^j$ is expanded to match image size and concatenated to channels of their corresponding sample image $j$ before the input block is applied. For all image experiments we follow the timestep $\gamma$ schedule used in De Bortoli et al. (2021) with $\gamma_{\min} = 10^{-5}$ and $\gamma_{\max} = 0.1$. We use Adam optimizer with learning rate $10^{-4}$ and momentum 0.9. Experiments with image datasets were run on limited shared GPU resources; lower-resolution image sizes and number of samples in cache were chosen accordingly.

**MNIST.** For the experiment in Figure 3, we use 10,000 cached images of size $28 \times 28$; the batch size is 128 and the number of timesteps is 30. The noise values are sampled uniformly in the

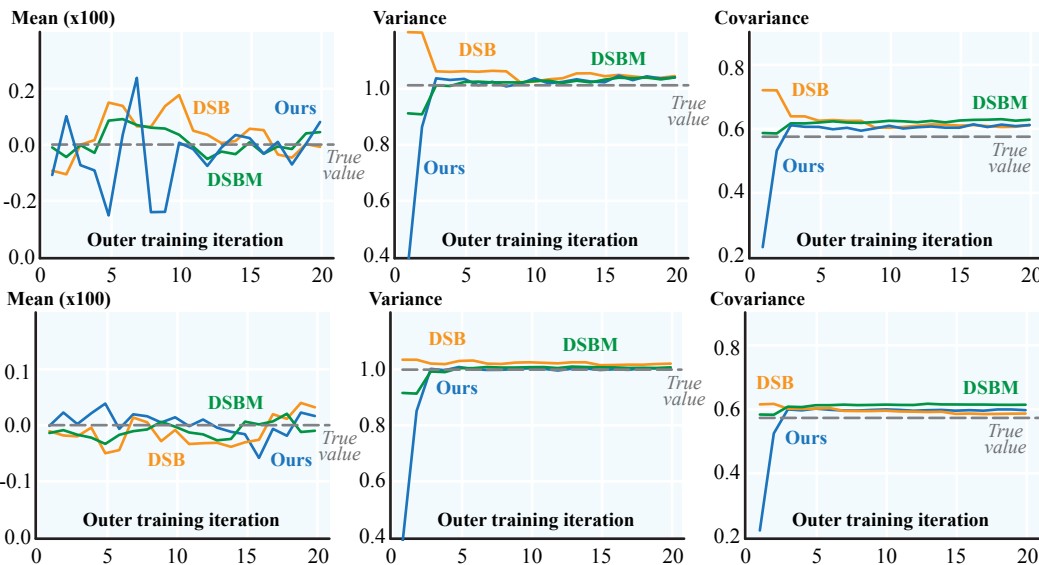

Figure 6: For each method, we plot the mean (left) and variance (middle) obtained for the terminal samples, i.e. samples obtained at time $t = T$, as well as the covariance (right) of the joint distribution, versus the number of outer iterations, averaged over 5 trials. Top: $d = 5$. Bottom: $d = 20$.

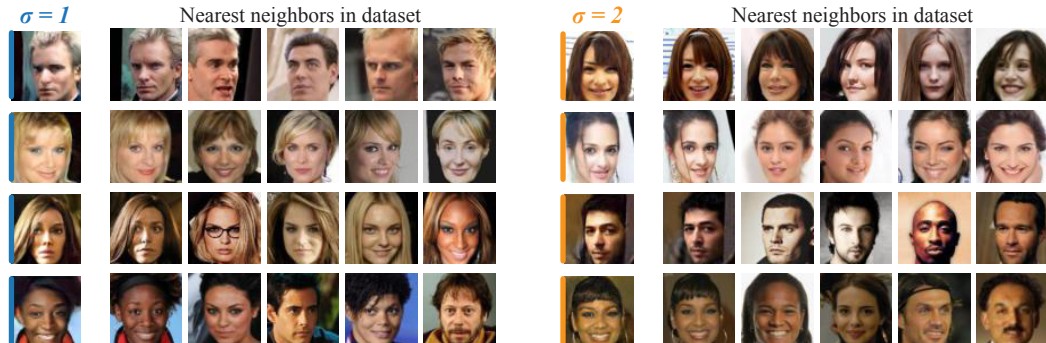

Figure 7: For our generated results (first and seventh columns), we show the five nearest neighbors in the CelebA dataset as measured through the features extracted by ResNet50 (He et al., 2016).

interval from 1 to 5 (inclusive) during training. We train for 5,000 iterations per outer iterations, and cached samples are refreshed every 1,000 inner iterations. The terminal samples shown are for outer iteration 8.

**CelebA.** In Figures 4 and 12, we use 300 cached images of size $64 \times 64$ and batch size 128. The cache is refreshed every 100 inner iterations and we train for 5,000 iterations per outer iterations. The number of timesteps is 50; the $\sigma$ values are uniformly sampled in the interval from 1 to 3. The terminal sample images are shown for outer iteration 15. The FID score in Figure 4 is computed using 300 images.

**Flowers102.** For Figure 5, we use 500 cached images of size $64 \times 64$. The batch size is 128 and cache is refreshed every 100 inner iterations. We train for 5,000 inner iterations per outer iteration. Terminal samples are shown for outer iteration 20. The $\sigma$ values are uniformly sampled in the interval from 1 to 5; the number of timesteps is 50.

# D ADDITIONAL EXPERIMENTAL RESULTS

## D.1 CONTROL OVER SAMPLE PROXIMITY

We define proximity of samples using pixel-wise $L_2$ norm as our choice of distance metric. In Figure 8 (left), we demonstrate how larger values of $\sigma$ effectively produce pushforward samples that are farther in this distance metric, compared to samples generated with smaller values of $\sigma$. This experiment expands the results shown in Figure 2 to the case of resampling from image distributions.

**A larger $\sigma$ produces more distant outputs...**   **...and takes more convoluted paths to get there**

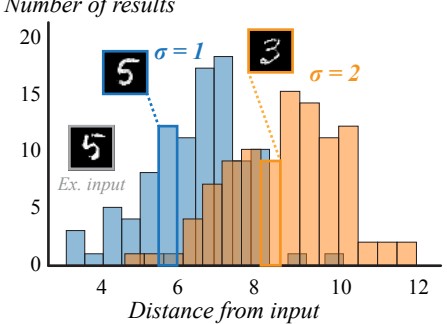
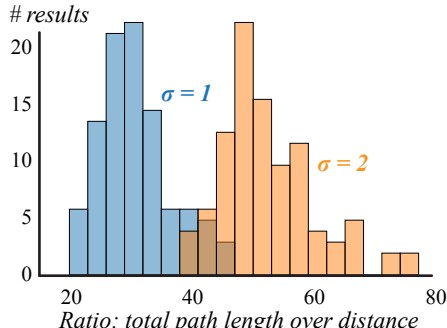

Figure 8: On the left: Two histograms demonstrating how image samples generated with larger $\sigma$ correspond to less proximal samples relative to the initial image sample. On the right: Two histograms show the inverse ratio between displacement and total path length of sample paths as a metric of path regularity.

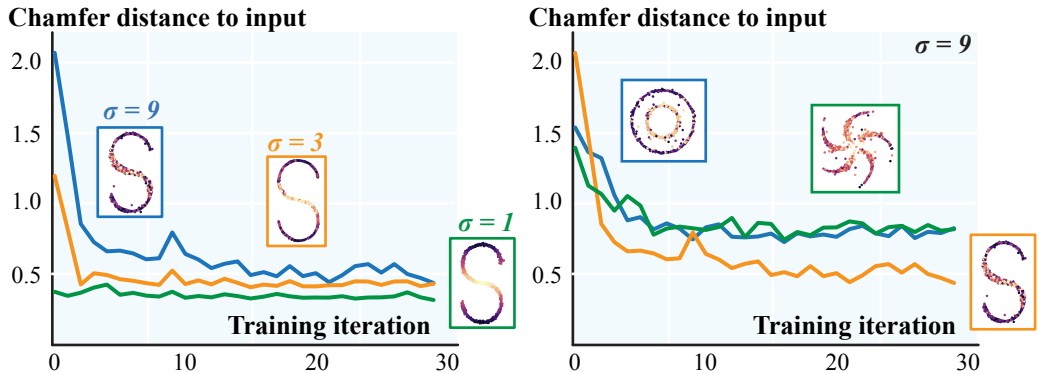

Figure 9: On the left: Three curves, each corresponding to a different $\sigma$ value, showing convergence using Chamfer distance for the same 2D dataset (shown in Figure 2). On the right: Three curves, each corresponding to a different 2D dataset, showing convergence for a fixed $\sigma$ value.

In particular, the mean and spread of the histograms in Figure 8 (right) show that larger values of sigma correspond to higher average distance values relative to the initial sample, as well as greater variation among these distances.

NEW

### D.2 SAMPLE PATH REGULARITY

We present empirical results on the regularity of path measures produced by our method. Specifically, in Figure 8 (right), we give a histogram for the values of a metric defined by taking the ratio of total displacement to total path length for different values of $\sigma$. For a given sample trajectory $\{\mathbf{X}_i\}_{i=0}^{M-1}$, this metric is explicitly computed by dividing $\|\mathbf{X}_0 - \mathbf{X}_{M-1}\|_2$ (total displacement) and $\sum_k \|\mathbf{X}_{k+1} - \mathbf{X}_k\|_2$ (total path length). The greater the value of this metric, the greater the variation in the trajectory; hence, smaller values of this metric are suggestive of greater sample path regularity. We find, as expected, that sample path regularity decreases as $\sigma$ increases.

NEW

### D.3 INTEGRITY OF INITIAL DISTRIBUTION

We compute Chamfer distances as a means of measuring the proximity of the pushforward distributions exhibited in Figure 2 to the corresponding initial distributions. In the mirror case, the pushforward distribution should match the initial distribution, and the Chamfer distance between them should therefore decay as the number of iterations grows. In Figure 9, we demonstrate how the Chamfer distance decays over outer iterations of our method for the same 2D distribution with different values of $\sigma$ (left), as well as how the Chamfer distance decays for different datasets with fixed $\sigma$ value (right).

NEW

### D.4 COMPARISON TO ALTERNATIVE METHODS

We compare our method with DSB and DSBM for image resampling with the MNIST dataset as the initial and final marginal distribution. For this experiment, we use the implementation for DSB and DSBM-IPF available in the code repository for Shi et al. (2023). We implement our algorithm based on the architecture provided, only modifying the model to take on $\sigma$ as an input parameter for our method. We test all three methods with the same set of training parameters as described in Appendix C.3. We train our model with $\sigma = 1$ fixed to match the noise value in the SDE for the other two methods, which do not take $\sigma$ as a model input.

We provide FID scores for each method in Figure 10. We observe that for DSB and DSBM, the forward and backward models result in pushforward samples of different quality. In particular, sample quality for the forward model is significantly lower than that of the backward. This indicates that neither of these methods converge to the mirror Schrödinger bridge for the given number of iterations, because the drift function for this bridge is necessarily time-symmetric, i.e., the forward and backward drifts must be equal to each other. In contrast, our algorithm provides time-symmetry

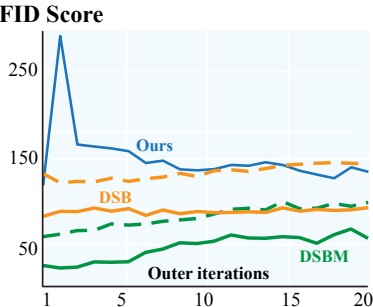

Figure 10: On the left: FID scores of pushforward samples versus outer iterations (single run) produced by our method, by DSB, and by DSBM, for a mirror bridge with the MNIST dataset as the marginal distribution. Solid lines correspond to backward models and dashed lines to forward models. On the right: Breakdown of runtimes at iteration 20 for the same experiment on each of the three methods.

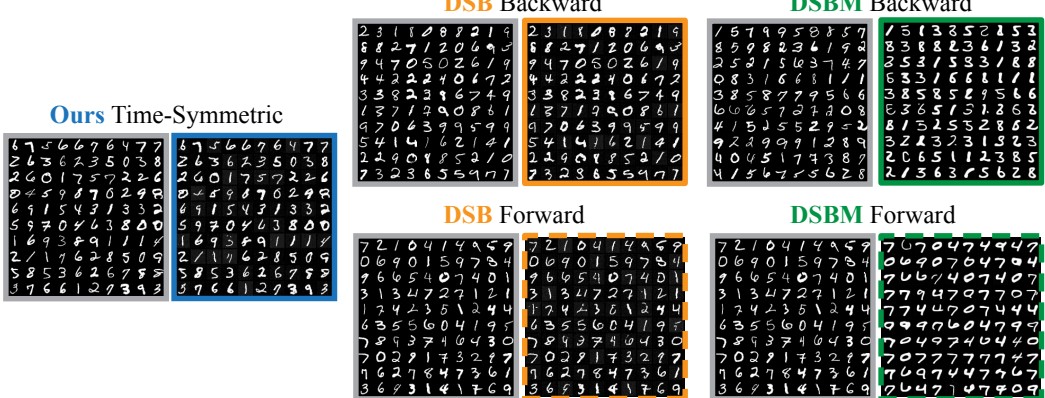

Figure 11: Result of image resampling at outer iteration 20 for the experiment in Figure 10. For each method and drift direction, the initial samples are displayed on the left and the pushforward samples on the right.

by construction: a single model is trained and forcibly "symmetrized" at each outer iteration via the drift averaging procedure described in Section 4.3.

Also in Figure 2, we present a breakdown of runtime for each method obtained for the same experiment. Our method has significantly lower total runtime and average outer training iteration time. The latter is not surprising, considering that one of the key features of our algorithm is to eliminate training for one of the projection steps taken; recall that we perform the reverse $D_{\mathrm{KL}}$ projection completely analytically. We observe that the average inference time during training, however, is higher with our method. Overall, in this particular experiment, we see that our method makes a trade-off between a small reduction in sample quality for a significant speed-up in training, while also preserving the time-symmetry of the solution.

NEW

## D.5  ADDITIONAL CELEBA RESULTS

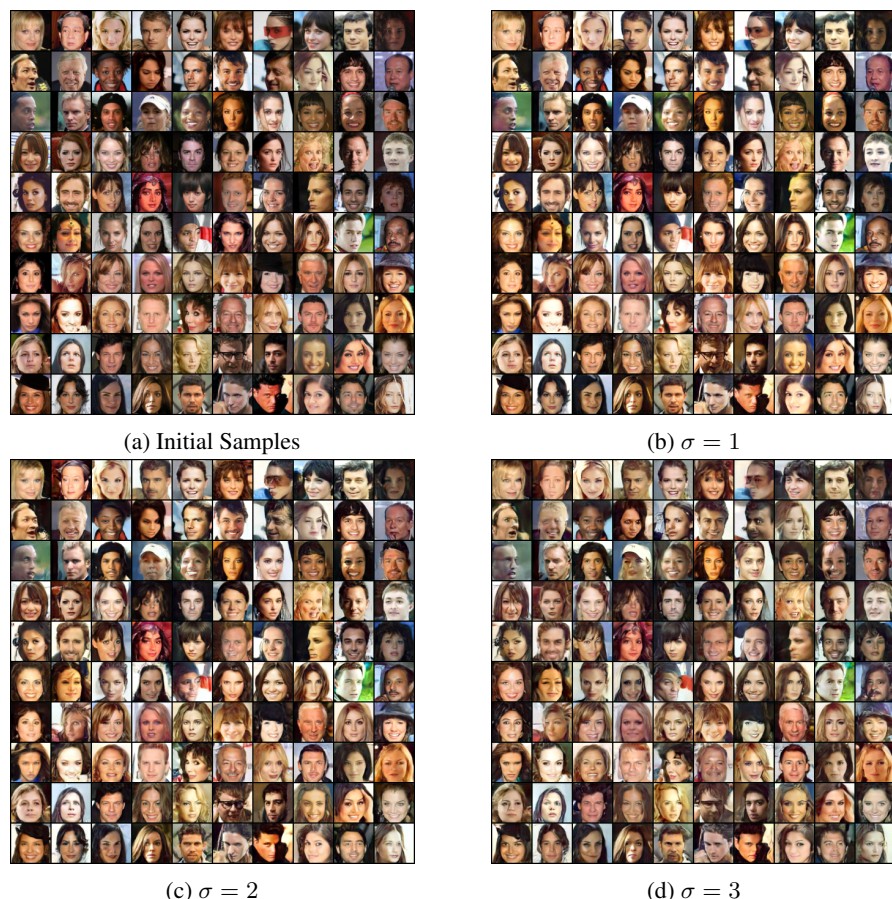

(a) Initial Samples

(b) $\sigma = 1$

(c) $\sigma = 2$

(d) $\sigma = 3$

Figure 12: Additional results for the empirical distribution of images in CelebA from which the examples in Figure 4 are obtained.

