# OpenReview forum: "Through the Looking Glass: Mirror Schrödinger Bridges"
_ICLR.cc/2025/Conference — Submitted to ICLR 2025_

### Official Review · Reviewer_yMTk · 2024-10-30

**Soundness:** 3
**Presentation:** 3
**Contribution:** 2
**Rating:** 6
**Confidence:** 2

**Summary:**

The authors consider a modification to the original Schrödinger bridge problem where they consider the marginals to be the empirical measure of the data. This learns a coupling between two sets of the same data. The coupling is designed to optimal in the relative entropy sense. They modify the iterative proportional fitting procedure such that they project in the direction that minimizes the KL divergence in one step and then in the reverse KL divergence in the next step due to the analytical feasibility of the step. They propose a practical algorithm for computing the projections using a change of measure technique and optimizing the drifts.

**Strengths:**

The authors provide an interesting perspective on the Schrödinger bridge problem and provide a new technique for fitting a path measure connecting an initial condition given by itself to itself.

The method is fairly straightforward to implement and say to analyze.

The method provides optimality with respect to relative entropy, which is a nice property of the sample paths.

**Weaknesses:**

My main concerns come with the empirical evaluation of the method.

While the method has nice motivation, empirically the results do not seem to be impressive. In the cases of low \sigma, the variation is very small compared to the initial condition. This is likely because the method is estimating a map between itself that is effectively an OU process. Furthermore, when the $\sigma$ is large, the methods appear to be much more corrupt and lose some of the important features.

In general the performance of the method does not seem to be well studied. Since one of the motivations the authors mentioned was based on the path measure being optimal with respect to relative entropy, I would show some of these results on the regularity of the path space.

**Questions:**

When training, how are the data split? Is there any pairing done between the data points? How would that affect the mapping between the two sets.

Are there other mean reverting processes that you can consider other than OU processes? How do these affect the performance of the method?

Is there a set of results one can consider that consist of analyzing the regularity of the path measures?

---

> ### Author Response · Authors · 2024-11-16
>
> We appreciate your feedback. Attached is a PDF that reflects changes addressing your concerns. Below, we respond to each of your comments and propose experiments that will be posted within several days. We believe that these address all concerns raised in your original review, but **please let us know as soon as possible if there are any other experiments or clarifications needed to strengthen our case**. We are confident that we can address these during the rebuttal period.
>
> > “My main concerns come with the empirical evaluation of the method. [...] In the cases of low $\sigma$, the variation is very small compared to the initial condition. This is likely because the method is estimating a map between itself that is effectively an OU process. Furthermore, when the σ is large, the methods appear to be much more corrupt and lose some of the important features.”
>
> We would like to point out that despite limited computational resources, our method still produces promising results. As evidence of this, we refer the reviewer to the results presented in De Bortoli (2021), specifically in Figure 4. Their results show significant image degradation, but subsequent papers showed higher resolution results when using significantly more computational resources. Similar to these works, we expect our method to produce more competitive results if given increased computational resources.
>
> > “When training, how are the data split? Is there any pairing done between the data points? How would that affect the mapping between the two sets.”
>
> There is no pairing done between data points. In regards to the training data, we split the train labeled portion of each dataset into shuffled batches. Training samples are refreshed after a fixed number of iterations. Further specification on batch size, number of samples, and other data splitting parameters for each example is included in our original submission in Appendix C Implementation Details. The portion of the dataset labeled as test is split into batches used at test time. There is no overlap between train and test data.
>
> > “Are there other mean reverting processes that you can consider other than OU processes? How do these affect the performance of the method?”
>
> Yes, there are probably other mean reverting processes that can be considered. The OU process is a natural candidate, but any time-symmetric process can be used. Not every mean reverting process, however, is time-symmetric, and time-symmetry of the prior is a requirement for the derivation of our method. The OU process is arguably the simplest case of a time-symmetric mean reverting process. This allows us to keep the method implementation straightforward at no cost to performance.
>
> > “Is there a set of results one can consider that consist of analyzing the regularity of the path measures?”
>
> This is an excellent question! We propose adding a time increments experiment within the next few days to show regularity of the path measures. Specifically, one way to analyze the regularity of the path measures is to examine the difference between samples at subsequent time steps. If these differences decrease with time step size, one can reasonably expect regularity of the path measures. Our image resampling data is particularly useful for this experiment since the timesteps are sampled on a schedule rather than uniformly. We will use this data to plot time increment versus time step size. We believe that this experiment and analysis should address this question as well as resolve the related concern raised in the weaknesses section of the review. **Please let us know if this experiment would suffice to clarify this question and strengthen our case.**

---

> ### Author Response · Authors · 2024-11-21
>
> We have posted an **updated version of the PDF including an experiment that addresses the one remaining concern in your review**, i.e. results on the regularity of path measures. Please review the addition of Appendix D2 Regularity on line 936 (labeled “NEW” in Green) and Figure 8. There, we show a metric of total path length over distance for different values of $\sigma$. For a given sample trajectory X_i, we obtain this metric via the inverse ratio of || X_0 - X_{M-1} || (distance) and $\sum$|| X_{k+1} - X_{k} || (total path). Smaller values of this metric loosely indicate regularity of the sample path. Please let us know as soon as possible if any further clarification or specific analysis is needed.
>
> **In light of the new additions, as well as the prior responses provided, we ask you to please consider raising your score.** We appreciate your feedback and are looking forward to receiving a follow up from you soon!

---

> ### Author Response · Authors · 2024-11-25
>
> **We hope that your review has been fully addressed with the latest PDF update and the previous responses** we provided in the official comments. **We ask that you please refer to these and consider raising your score.** We greatly appreciate the improvements made to our paper thanks to your review.
>
> In specific, we have addressed your comments by expanding the empirical evaluation of our method (see Appendix D on pages 17-19) with an experiment indicating regularity of sample paths and a comparison with alternative methods for image resampling. We have also provided direct answers to questions raised in your review. Please refer to the updated PDF and the previous comments in this thread for further details. If any further comments or questions arise please let us know as soon as possible as the review period is coming to an end.

---

> > ### Comment · Reviewer_yMTk · 2024-11-26
> >
> > Thank you for your rebuttal and apologies for the delayed response. I still have concerns regarding the empirical evaluation as I discussed in the initial review, even after considering the updated evaluation. Specifically, I am still concerned about the resampling quality even after consulting De Bortoli (2021), I would be curious in seeing alternative reference measures and if there are other relatively efficient processes that could be used improve this quality in followup work. I have adjusted my score accordingly.

---

> > > ### Author Response · Authors · 2024-11-26
> > >
> > > Thank you for your response. We very much appreciate you taking the time to review our modifications and we share your curiosity about this direction for future follow-up work. Thank you for raising your score.

---

### Official Review · Reviewer_Rjzk · 2024-11-02

**Soundness:** 2
**Presentation:** 2
**Contribution:** 2
**Rating:** 6
**Confidence:** 4

**Summary:**

This work studies the Schrödinger bridge (SB) between a distribution and itself, unlike most of the literature which focuses on the SB between two distributions (e.g., the standard Gaussian, and the data distribution). The authors propose this model as a means for conditional resampling of a distribution, where the "noise" induced by the bridge process allows them to obtain new samples which are in-distribution. They demonstrate their approach on many experiments, and have some proofs of their technical results.

**Strengths:**

This paper proposes to alleviate some computation burden of training SBs by proposing a learning algorithm that learns the SB between a measure and itself. Still faithful to the generative modeling paradigm, they learn the SB from a data distribution to itself, with the goal of starting at existing samples and generating diverse ones. The writing is relatively clear, and Figure 2 is especially clear at describing the phenomenon of "starting from an existing sample" and, given enough noise, learns to go somewhere else in the distribution.

**Weaknesses:**

While the computational burden of having to train two neural networks instead of one is appealing, there is little comparison between MSB and DSB or DSBM in terms of quantifying sample versatility (they performed some comparisons in the Gaussian case, but these are far from conclusive). For instance, in Figure 3: is there a certain $\sigma_\star$ after which the data generated by the MSB changes classes? This is likely hard to prove theoretically, but knowing if there exists some threshold after which data stops being the same class would be very interesting.

**Questions:**

Comments:
- Line 053: maybe write what "\delta"-measure means (or just say Dirac measure).
- The paragraph just above figure 3 is very unclear. I'm not sure what the rows and columns are meant to refer to here..
- The ability to use one neural network is not surprising from the connection between EOT and the SB problem. Equation (24-25) in the work by Feydy et al. (2019) precisely proposes some kind of fixed-point equation on one potential function (whereas for entropic OT between two measures, there are typically two potentials to optimize over via the Sinkhorn algorithm)
- Question: Is there any hope to provide a *rule* for choosing the amount of noise added in the process of generating images?
- Question: The choice of OU process appears entirely arbitrary. Why not consider standard Brownian motion as the reference process?

---

> ### Author Response · Authors · 2024-11-16
>
> We appreciate your thoughtful comments and suggestions! We look forward to improving our submission based on your feedback. Attached is a PDF that reflects changes addressing your concerns. Below, we respond to each of your comments and propose experiments that will be posted within the next few days. We believe that these address all concerns raised in your original review, but **please let us know as soon as possible if there are any other experiments or clarifications needed to strengthen our case**. We are happy to follow up with you and confident that we can address all of your concerns within the rebuttal period.
>
> > “[...] there is little comparison between MSB and DSB or DSBM in terms of quantifying sample versatility (they performed some comparisons in the Gaussian case, but these are far from conclusive).”
>
> We strongly agree with your suggestion! We propose adding a comparison to baseline methods for image resampling. Specifically, we will post an experiment within several days that compares results in Section 5 for 2D Datasets and Image Resampling to other methods, such as DSB or DSBM. **Please let us know if this would suffice to address this concern.**
>
> > “[...] in Figure 3: is there a certain σ⋆ after which the data generated by the MSB changes classes? This is likely hard to prove theoretically, but knowing if there exists some threshold after which data stops being the same class would be very interesting.”
>
> This is a very interesting question! We added this to our conclusion in Section 6 Lines 538-539 as a direction for future research. This is a hard question from both a theoretical and empirical standpoint. For instance, consider the MNIST classes. Our method essentially flows samples out of the manifold and back to it; the new sample “lands” at a distance proportional to the noise value. A threshold for changing class, however, would depend not only on this relationship between noise and spread of data, but also on the location of the initial sample relative to the boundary between classes in the data manifold.
>
> > “Line 053: maybe write what "\delta"-measure means (or just say Dirac measure). The paragraph just above figure 3 is very unclear. I'm not sure what the rows and columns are meant to refer to here..”
>
> We have modified our original submission to clarify these two concerns. We believe these are fully addressed in the current version of the PDF. Please refer to the text added in Orange (labeled “FIX”) in Line 53 (Introduction) and in Lines 508-510 (Section 5 for Image Resampling). Thank you for bringing this to our attention!
>
> > “Equation (24-25) in the work by Feydy et al. (2019) precisely proposes some kind of fixed-point equation on one potential function (whereas for entropic OT between two measures, there are typically two potentials to optimize over via the Sinkhorn algorithm)”
>
> We have added a comment in Section 4 Lines 193-196 to highlight this connection. Our original submission cites Feydy (2019) in Section 2, but we agree that this connection can be more clearly highlighted to readers when discussing the single drift function in our method. We would like to point out that Feydy (2019) considers leveraging the symmetry of the transport problem in the static case. When considering the SB problem, an approach needs to be developed for the dynamical formulation in the language of path measures, which differs from theirs (and, similarly, from that of Kurras (2015)). This is one of the main contributions of our work. We believe the addition made in the PDF addresses this comment. Thank you for bringing this up!
>
> > “Is there any hope to provide a rule for choosing the amount of noise added in the process of generating images?”
>
> There seems to be little hope of providing a rigorous rule, but our experiments suggest some general patterns. The relationship between the noise value and the spread of samples we presented for the Gaussian case in Section 4.4 provides some intuition as to why it is not possible to establish a general rule. In particular, the spread of samples in the Euclidean sense is directly proportional to the value of sigma. The constant of proportionality depends on variable parameters, including choice of dataset, choice of initial drift, and number of timesteps.
>
> > “The choice of OU process appears entirely arbitrary. Why not consider standard Brownian motion as the reference process?”
>
> This is a theoretical requirement. The prior needs to be a time-symmetric measure for the bridge to be time-symmetric. Standard Brownian motion is not time-symmetric, and the OU process was chosen for this reason. Time-symmetry is one of the key properties that the derivation of our algorithm hinges on.
> We appreciate you bringing this to our attention and we have highlighted this requirement in Section 4.1 Lines 221-222 to clarify its importance to readers. Further theoretical details can be found in Agarwal (2024), which is cited in our original submission.

---

> ### Author Response · Authors · 2024-11-22
>
> One more experiment comparing our method to alternatives in the image resampling case will be posted in the next day or two. We will keep you posted, as soon as it is available, but would appreciate a follow up in the meantime. **Please let us know as soon as possible if there are any other clarifications needed based on the responses we already provided, or if these items are fully addressed.** We appreciate your feedback and are looking forward to hearing back from you soon! **In light of the new additions, which address questions in your review, as well as the prior responses provided, we ask you to please consider raising your score, especially once your review is fully addressed.**

---

> ### Author Response · Authors · 2024-11-24
>
> We have posted **a new PDF update that we believe fully addresses the concerns in your review**. In particular, please refer to Appendix D where we substantially expand on the empirical evaluation of our method. In the newly added Appendix D4, and Figures 10 and 11, you will find a comparison of our method with DSB and DSBM for image resampling. For convenience, we have summarized the conclusions draw from the comparison experiment here:
>
> Dataset: MNIST. Task: Image resampling.
>
> | **Runtime** | Ours | DSB | DSBM-IPF |
> |---|---|---|---|
> |Total | **2.64hrs** | 5.25hrs   | 12.47hrs  |
> | Avg. Outer Iter. |  **7.94min**  |  15.7min  | 37.41min  |
> | Avg. Inner Iter. | 0.059s   |  **0.055s**  |  0.209s |
> |Avg. Inference |  2.009s  |  1.554s  |  **1.002s**  |
>
> FID (Iteration 20) | Ours | DSB | DSBM-IPF |
> |---|---|---|---|
> |   | 135.4  | N/A | N/A  |
> | Backward model|  N/A |  93.65  |  56.32 |
> | Forward model | N/A | 144  |  98.89 |
>
> **Overall, in this particular experiment, we see that our method makes a trade-off between a small reduction in sample quality for a significant speed-up in training, while also preserving the time-symmetry of the solution.**
>
> We have also tested the same experiment using the CelebA dataset and observed issues with mode collapse with one of the alternative methods. Given the short discussion window and to be fair to the method in question, we have omitted these results but would like to run more ablation tests to possibly include these for the camera-ready version of our paper.
>
> **We believe that your review has now been fully addressed with the new additions and ask that you please consider raising your score.** We are thankful for the suggestions and comments made in your review, and we believe our current version of the paper has been much improved by addressing them!

---

> ### Author Response · Authors · 2024-11-25
>
> **We hope that your review has been fully addressed with the latest PDF update and the previous responses** we provided in the official comments. **We ask that you please refer to these and consider raising your score.** We greatly appreciate the improvements made to our paper thanks to your review.
>
> In specific, we have addressed your comments by providing several new experiments (see Appendix D on pages 17-19) such as comparison with alternative methods for image resampling, and answers to all the questions in your initial review. Please refer to the updated PDF and the previous comments in this thread for further details. If any further comments or questions arise please let us know as soon as possible as the review period is coming to an end.

---

> > ### Comment · Reviewer_Rjzk · 2024-11-25
> > **Thank you**
> >
> > Thank you for all your efforts; I will take a careful look at the updated draft and consider updating my score by the end of the period.

---

### Official Review · Reviewer_gCiH · 2024-11-04

**Soundness:** 4
**Presentation:** 4
**Contribution:** 3
**Rating:** 5
**Confidence:** 3

**Summary:**

The paper introduces the mirror Schrödinger bridge, a method for addressing the resampling problem when the initial and target distributions are identical. Unlike traditional Schrödinger bridges, which are designed for mapping between two distinct distributions, the mirror Schrödinger bridge is formulated specifically for self-mapping within a single distribution. This unique approach facilitates the generation of in-distribution variations of data points, allowing for conditional resampling that maintains the original distribution’s integrity.

The authors develop a theoretical foundation for this method, employing time symmetry and the Alternating Minimization Procedure to establish convergence in the total variation metric, even for infinite-dimensional state spaces. This achievement addresses the challenging issue of convergence in high-dimensional settings. Additionally, the algorithm capitalizes on the time symmetry inherent in the problem, enabling it to model the diffusion drift with a single neural network. This innovation significantly reduces computational costs, effectively halving the effort compared to Iterative Proportional Fitting Procedure based approaches.

Empirical evaluations underscore the practical value of the mirror Schrödinger bridge across diverse applications, highlighting its capability to produce high-quality proximal samples that are valuable for tasks like data augmentation and generative modeling. In summary, this research claims to provide a theoretically rigorous and computationally efficient solution for conditional resampling within the same distribution, combining solid theoretical contributions with practical algorithmic advancements.

**Strengths:**

The paper introduces the mirror Schrödinger bridge framework, an approach that differs from traditional Schrödinger bridges by focusing on mapping a distribution onto itself rather than between distinct distributions. This self-mapping approach directly addresses the challenge of conditional resampling within the same distribution, opening up new possibilities for generating in-distribution variations of data points. By incorporating time symmetry and the Alternating Minimization Procedure (AMP) to establish theoretical foundations, the paper presents an innovative solution to resampling. The algorithm also leverages time symmetry to train a single neural network for modeling the diffusion process drift, enhancing computational efficiency.

The paper includes solid theoretical foundations and provides comprehensive convergence proofs in the total variation metric, even in infinite-dimensional state spaces. The AMP is carefully developed and shown to converge to the mirror Schrödinger bridge, ensuring methodological consistency. The algorithm’s implementation is efficient, theoretically reducing computational overhead by half compared to Iterative Proportional Fitting Procedure (IPFP)-based methods. Empirical evaluations across several applications support some theoretical claims, demonstrating the method’s capability to generate high-quality proximal samples for tasks like data augmentation and generative modeling. The organized presentation of theoretical and empirical findings underscores the paper’s contribution to the field.

Additionally, the paper is well-written and structured, guiding the reader through complex concepts with clarity. The transition from theoretical foundations to practical algorithmic details is seamless, ensuring a coherent flow. Most definitions and problem formulations are clearly presented. Explanations of AMP and iterative schemes enhance understanding, while algorithmic pseudocode supports practical comprehension.

**Weaknesses:**

While the paper introduces mirror Schrödinger bridges, the framework retains substantial mathematical similarities to traditional Schrödinger bridges. Could you clarify the key mathematical distinctions between the two approaches? The theoretical analysis would benefit from an error analysis for the Alternating Minimization Procedure (AMP) outlined in equations (4) and (5), which would provide valuable insights into the convergence rate of the proposed approach. The absence of such an analysis limits our understanding of the efficiency and accuracy of the AMP.



On the practical side, the algorithm does not include comparisons with other established approaches in the literature, nor is there any discussion regarding the impact or choice of specific discretization methods, such as the Euler-Maruyama scheme, in addressing this problem. Furthermore, the paper lacks a detailed explanation of all the Figures, which hinders the reader's ability to assess how the proposed method performs relative to existing methods and to understand the extent to which halving the iterations reduces runtime quantitatively.

The paper also lacks a quantitative flow for evaluating the equality of resampling across examples and fails to specify metrics used to assess performance. An empirical definition of "proximity" would add clarity. For instance, it is unclear whether a proximity value of 5 remains constant with $\sigma=1$ or changes to 3 with a different $\sigma$, and the way proximity is quantified in empirical examples remains vague. Although the paper claims validation across various application domains and provides some information on the experimental setup and datasets, it lacks sufficient detail on the specific metrics used for performance assessment, making it difficult to evaluate the method's effectiveness and robustness relative to existing techniques.

Finally, while the paper emphasizes algorithmic simplifications that reduce computational costs by training only one neural network, it does not address the scalability of the method to high-dimensional data. The absence of any analysis on how the method performs as the data dimensionality increases leaves questions about its applicability to high-dimensional settings, which are increasingly relevant in practical applications.

**Questions:**

1. **Error Analysis for AMP**: The paper introduces the Alternating Minimization Procedure (AMP) in Equations (4) and (5) but lacks an error analysis. Could you elaborate on the convergence speed of AMP? Specifically, are there theoretical bounds or guarantees on the convergence rate that would enhance the theoretical foundation of your method?

2. **Benchmarking Against Existing Methods**: Could you clarify which algorithms you compared with your proposed method for each example case in Section 5? Benchmarking against established methods would help situate your approach within existing literature.

3. **Choice of Euler-Maruyama**: Your algorithm employs Euler-Maruyama discretization. How does this choice impact the accuracy and efficiency of solving the Schrödinger bridge problem? Have you considered alternative discretization schemes, and how do they compare in terms of performance?

4. **Iterations vs. Runtime**: How does halving the number of iterations quantitatively affect running time? A breakdown of runtime reductions relative to iteration count would provide a clearer picture of the efficiency gains.

5. **Quantitative Metrics and Proximity Definition**:
   - **Metrics for Resampling Quality**: The paper lacks specific metrics to assess resampling quality in each example case. What metrics do you use to evaluate how well the method preserves the integrity of the original distribution during resampling?
   - **Proximity Definition**: An empirical definition of proximity would clarify how closely generated samples align with input data. For example, on MNIST, how would a proximity value of a digit-5 image with \(\sigma=1\) compare to that of a digit-3 image with a different \(\sigma\)?

6. **Experimental Setup Details**: Although the paper claims empirical validation across various domains, the experimental setup lacks specificity. Could you provide more details on the datasets, experimental procedures, and metrics used to assess performance?

7. **Quantitative Results and Comparisons**: Could you include performance metrics and comparisons that demonstrate the robustness and potential advantages of mirror Schrödinger bridges over existing techniques?

8. **Scalability to High-Dimensional Data**:
   - **Performance on High-Dimensional Benchmarks**: The scalability of your method to high-dimensional data is not discussed. Could you provide insights into its performance and computational complexity on high-dimensional or large-scale datasets?
   - **Strategies for High-Dimensional Data**: What strategies do you propose to ensure the scalability and efficiency of mirror Schrödinger bridges when applied to high-dimensional data or data on manifolds? Are there modifications or optimizations that could improve performance in these scenarios?

9. **Interpretation of Figure 2**: In Figure 2, the resampling estimate with $\sigma=1$ appears to produce more concentrated samples compared to the original samples. Could you explain this behavior? Is there some form of geometric or manifold enhancement occurring in the resampling process?

---

> ### Author Response · Authors · 2024-11-16
>
> Response Thread (1/2). We thank you for your valuable feedback. Below, we respond to each comment, along with proposed additional experiments that we will post within the next few days. Please also refer to the attached PDF reflecting these changes.
>
> **If there are further experiments or concerns that require clarification, please let us know as soon as possible.** We are committed to improving our paper and are confident in our ability to address your concerns within the rebuttal period.
>
> > “[...] the framework retains substantial mathematical similarities to traditional Schrödinger bridges. Could you clarify the key mathematical distinctions between the two approaches?”
>
> Thank you for highlighting this point. While our method is clearly linked to previous approaches for learning the Schrödinger bridge, such as DSB, our key mathematical insight is that in the case of identical marginals, we can leverage time-symmetry to derive a different projection-type algorithm. While the outer structure of our algorithm retains two projection steps, as is the case with DSB, the projection steps themselves are different. In particular, one of our projection steps is completely analytic and requires no learning to perform. The latter allows us to perform an algorithm with a single variable (instead of two) and obtain a solution more efficiently (with half the training iterations).
>
> > “The paper introduces the Alternating Minimization Procedure (AMP) in Equations (4) and (5) but lacks an error analysis. Could you elaborate on the convergence speed of AMP? Specifically, are there theoretical bounds or guarantees on the convergence rate that would enhance the theoretical foundation of your method?”
>
> This is an interesting question, thank you for asking! The convergence rate can be readily obtained from equation (6) in the paper. In short, the convergence rate is o(1/n) where n is the number of iterates. We have added this analysis to the proof (and accompanying statement) of Theorem 1 in Section 4.2 Lines 287-290.
>
> > “Could you clarify which algorithms you compared with your proposed method for each example case in Section 5? Benchmarking against established methods would help situate your approach within existing literature.”
>
> Section 5 compares our method to DSB and MSBM, two algorithms used to learn general Schrödinger bridges. The comparison in the original submission is done for the Gaussian transport case; it can be found in Figure 1.
>
> We agree that further comparison will improve our submission and we appreciate your suggestion. We propose an experiment, which we will post within several days, that compares results in Section 5 for 2D Datasets and Image Resampling  to other methods for the image resampling examples. Please let us know if this experiment would suffice to address this comment.
>
> > “Your algorithm employs Euler-Maruyama discretization. How does this choice impact the accuracy and efficiency of solving the Schrödinger bridge problem? Have you considered alternative discretization schemes, and how do they compare in terms of performance?”
>
> Euler-Maruyama (EM) allows us to compute the reverse drift as a regression problem. This approach has the advantage of approximating the drift without the need of computing the score function. Another advantage is that its implementation is straightforward. EM is the standard choice in the Schrödinger bridge and diffusion-based sampling communities for SDE integration.
>
> This choice of discretization in principle might impact accuracy and efficiency in the following ways:
> Since the computation is done via local estimates, the training process can suffer from divergences. This is addressed in De Bortoli (2021) by the implementation of exponential moving averages during training, and the same solution is implemented in our training framework. We have highlighted this in the revised PDF in Appendix C Implementation Details (please refer to the text added in Green labeled “NEW” in Lines 808-809).
>
> There is a tradeoff between computational expense and accuracy. The drift estimates are done locally using finite difference time intervals. Hence, the smaller the time intervals, the better the approximation is. On the other hand, sampling at more time intervals increases the computational cost.
>
> These are common disadvantages shared by alternative frameworks relying on EM discretization, such as De Bortoli (2021), Vargas (2021), and Winkler (2023). A broad study comparing SDE integrators’ effects on performance of Schrödinger bridge and diffusion models is an interesting topic for future work but outside the scope of our current study.
>
> **Please refer to the next Comment, where our response continues.**

---

> ### Author Response · Authors · 2024-11-16
>
> Response Thread (2/2).
>
> > “How does halving the number of iterations quantitatively affect running time? A breakdown of runtime reductions relative to iteration count would provide a clearer picture of the efficiency gains.”
>
> We propose adding runtimes to the convergence experiment in Figure 1. In addition, the comparison experiment we propose to post within several days should further address this concern for the image resampling case. Please let us know if these additions would fully address your concern.
>
> In principle, for a fixed value of sigma and identical marginal constraints, a single inner iteration of our method should not take longer than a single inner iteration of alternative IPFP-based methods, like DSB.
>
> > “The paper lacks specific metrics to assess resampling quality in each example case. What metrics do you use to evaluate how well the method preserves the integrity of the original distribution during resampling?”
>
> The typical metric to assess resampling quality for the image generation case is the FID score. This information is already included in our original submission in Figure 4 and we have added an explanation in Section 5 for Image Resampling Lines 510-512 to make it clearer to readers.
>
> Moreover, we are proposing to do an experiment where we run competing algorithms (e.g., DSB) to address another comment and we will report the FID scores for those. We believe that this will further address this comment.
> For the 2D datasets, we computed Chamfer distances as an in-distribution metric for the generated samples. We will add a table with these results to address this concern for the 2D examples.
>
> > “An empirical definition of proximity would clarify how closely generated samples align with input data. For example, on MNIST, how would a proximity value of a digit-5 image with ($\sigma=1$) compare to that of a digit-3 image with a different ($\sigma$)?”
>
> We agree that this would add clarity to the notion of proximity. We propose to add a table with the distance measured using a similarity metric between the initial sample and the two generated samples with different values of sigma and will follow up soon with this table included in our revision.
>
> > “Although the paper claims empirical validation across various domains, the experimental setup lacks specificity. Could you provide more details on the datasets, experimental procedures, and metrics used to assess performance?”
>
> We would like to clarify that details on the experimental setup are included in our original submission in Appendix C Implementation Details. In addition, our response to a prior question about metrics should clarify the latter half of this question, i.e., which metrics have been included in the original submission and which metrics we have added to the revision.
>
> > “In Figure 2, the resampling estimate with $\sigma=1$ appears to produce more concentrated samples compared to the original samples. Could you explain this behavior? Is there some form of geometric or manifold enhancement occurring in the resampling process?”
>
> That’s an excellent question! To gain intuition about this phenomenon, we can look at our results for Gaussian transport in Figure 1. While our method approaches the mean of the closed-form solution with small noise, it initially approaches the ground-truth variance and then moves slightly below it as the iterations increase. We expect that the higher concentration of samples you noticed in the lower-dimensional case with classical distributions in Figure 2 is a result of the same phenomenon, which was incidentally also observed in De Bortoli (2021).

---

> ### Author Response · Authors · 2024-11-21
>
> We have posted an **updated version of the PDF including new experiments that address concerns in your review**. Please review the addition of Appendix D1 Proximity on line 895 (labeled “NEW” in Green) and Figure 8. There, we show an empirical measure of distances and demonstrate how sigma in fact is correlated with this distance metric in the image resampling examples we showed earlier in Figure 2.
>
> Moreover, we ask that you please review the addition of Appendix D3 Integrity of Initial Distributions on line 943  (labeled “NEW” in Green) and Figure 9, where we present results using Chamfer distance as an in-distribution metric for the pushforward samples of 2D datasets to show that our method preserves the integrity of the original distribution for this case. We believe that this addresses concerns in your original review.
>
> One more experiment comparing our method to alternatives in the image resampling case will be posted in the next day or two. But **in light of the new additions, which address questions in your review, as well as the prior responses provided, we ask you to please consider raising your score**. We will keep you posted on the remaining addition, as soon as it is available, and would appreciate a follow up in the meantime. We appreciate your feedback and are looking forward to receiving a follow up from you soon!

---

> ### Author Response · Authors · 2024-11-24
>
> We have posted **a new PDF update that we believe fully addresses the concerns in your review**. In particular, please refer to Appendix D where we substantially expand on the empirical evaluation of our method. In the newly added Appendix D4, and Figures 10 and 11, you will find a comparison of our method with DSB and DSBM for image resampling. For convenience, we have summarized the conclusions draw from the comparison experiment here:
>
> Dataset: MNIST. Task: Image resampling.
>
> | **Runtime** | Ours | DSB | DSBM-IPF |
> |---|---|---|---|
> |Total | **2.64hrs** | 5.25hrs   | 12.47hrs  |
> | Avg. Outer Iter. |  **7.94min**  |  15.7min  | 37.41min  |
> | Avg. Inner Iter. | 0.059s   |  **0.055s**  |  0.209s |
> |Avg. Inference |  2.009s  |  1.554s  |  **1.002s**  |
>
> FID (Iteration 20) | Ours | DSB | DSBM-IPF |
> |---|---|---|---|
> |   | 135.4  | N/A | N/A  |
> | Backward model|  N/A |  93.65  |  56.32 |
> | Forward model | N/A | 144  |  98.89 |
>
> **Overall, in this particular experiment, we see that our method makes a trade-off between a small reduction in sample quality for a significant speed-up in training, while also preserving the time-symmetry of the solution.**
>
> We have also tested the same experiment using the CelebA dataset and observed issues with mode collapse with one of the alternative methods. Given the short discussion window and to be fair to the method in question, we have omitted these results but would like to run more ablation tests to possibly include these for the camera-ready version of our paper.
>
> **We believe that your review has now been fully addressed with the new additions and ask that you please consider raising your score.** We are thankful for the suggestions and comments made in your review, and we believe our current version of the paper has been much improved by addressing them!

---

> ### Author Response · Authors · 2024-11-25
>
> **We hope that your review has been fully addressed with the latest PDF update and the previous responses** we provided in the official comments. **We ask that you please refer to these and consider raising your score.** We greatly appreciate the improvements made to our paper thanks to your review.
>
> In specific, we have addressed your comments by providing several new experiments such as a study of control over proximity, plots demonstrating integrity of initial distribution, and benchmarking against existing methods for image resampling (see Appendix D on pages 17-19). In addition to these, we have also provided answers to questions in the review, and made modifications in the text to reflect these. Please refer to the updated PDF and the previous comments in this thread for further details. If any further comments or questions arise please let us know as soon as possible as the review period is coming to an end.

---

> > ### Comment · Reviewer_gCiH · 2024-11-25
> > **Reviewer's response**
> >
> > Thank you for your time and effort. I will thoroughly review the updated draft and take it into consideration before finalizing my score at the end of the review period.

---

> > > ### Comment · Reviewer_gCiH · 2024-12-02
> > >
> > > Dear Authors,
> > >
> > > Thank you for your thoughtful responses to my concerns and questions. After carefully reviewing your clarifications, I have decided to maintain my original score.
> > >
> > > I appreciate the effort you have put into addressing the issues. All the best.
> > >
> > > Sincerely,
> > >
> > > Reviewer gCiH

---

### Official Review · Reviewer_vpEm · 2024-11-04

**Soundness:** 3
**Presentation:** 3
**Contribution:** 3
**Rating:** 6
**Confidence:** 3

**Summary:**

This paper proposes mirror Schrodinger bridge (MSB), a model for conditional resampling. An alternating minimization procedure is used to solve for the MSB with a theoretical guarantee. On the empirical side, the MSB method is implemented to sample from both toy distributions and image distribution from the real world.

**Strengths:**

By using time-symmetry, MSB only requires a single neural network and half of the computational expense compared to other IPFP-based algorithms.

**Weaknesses:**

1. The theoretical results are limited to asymptotic analysis. The convergence rate is not presented;
2. In the empirical evaluation, comparison to baseline methods is limited to the Gaussian example in Section 5.1. As a result, it's not clear how MSB compares to other methods in real-world image generation.

**Questions:**

What's the connection and difference between the MSB method and the score-matching strategy (like Song et al. 2021)? What's the performance difference?

---

> ### Author Response · Authors · 2024-11-16
>
> We appreciate your thoughtful feedback and suggestions. We are eager to improve our paper and confident that we can address your concerns within the rebuttal period. Below, we respond to each comment and, in some cases, propose to conduct additional experiments within the next few days. We believe these experiments will fully address any remaining concerns. Please also refer to the **attached PDF** reflecting these changes.
>
> **Should there be any further experiments that would strengthen our submission or any further concerns left unaddressed, please let us know as soon as possible.**
>
> > “The theoretical results are limited to asymptotic analysis. The convergence rate is not presented.”
>
> We agree that this additional analysis would benefit our theoretical results and we appreciate the suggestion. In fact, the convergence rate can be readily obtained from equation (6) in the paper with little further analysis. We have added this result in the proof (and accompanying statement) of Theorem 1 in Section 4.2 Lines 287-290. In brief, the convergence rate is o(1/n), where n is the number of iterates.
>
> > “In the empirical evaluation, comparison to baseline methods is limited to the Gaussian example in Section 5.1. As a result, it's not clear how MSB compares to other methods in real-world image generation.”
>
> This is an excellent suggestion. We propose to add a comparison to baseline methods for image generation. Specifically, we will post an experiment within several days that compares results in Section 5 for 2D Datasets and Image Resampling to other methods, such as DSB or DSBM. Please let us know if this would suffice to address your concerns about the empirical evaluation.
>
> >“What's the connection and difference between the MSB method and the score-matching strategy (like Song et al. 2021)? What's the performance difference?”
>
> Thank you for bringing attention to this point. We have added a discussion on the connection and difference between MSB and score-based generative modeling (SGM) in Section 2 Lines 109-113. There, we point out that unlike SGMs, our method provides a tool to flow an existing sample somewhere else in the same data distribution with control over the spread of the newly obtained sample. In contrast, SGMs flow samples from a Gaussian to the data distribution. While a direct empirical comparison between the two is not appropriate, since these are two fundamentally different problem statements, we agree that this additional discussion improves the paper.

---

> ### Author Response · Authors · 2024-11-22
>
> One more experiment comparing our method to alternatives in the image resampling case will be posted in the next day or two. We will keep you posted, as soon as it is available, but would appreciate a follow up in the meantime. **Please let us know as soon as possible if there are any other clarifications needed based on the responses we already provided, or if these items are fully addressed.** We appreciate your feedback and are looking forward to hearing back from you soon! **In light of the new additions, which address questions in your review, as well as the prior responses provided, we ask you to please consider raising your score, especially once your review is fully addressed.**

---

> ### Author Response · Authors · 2024-11-24
>
> We have posted **a new PDF update that we believe fully addresses the concerns in your review**. In particular, please refer to Appendix D where we substantially expand on the empirical evaluation of our method. In the newly added Appendix D4, and Figures 10 and 11, you will find a comparison of our method with DSB and DSBM for image resampling. For convenience, we have summarized the conclusions draw from the comparison experiment here:
>
> Dataset: MNIST. Task: Image resampling.
>
> | **Runtime** | Ours | DSB | DSBM-IPF |
> |---|---|---|---|
> |Total | **2.64hrs** | 5.25hrs   | 12.47hrs  |
> | Avg. Outer Iter. |  **7.94min**  |  15.7min  | 37.41min  |
> | Avg. Inner Iter. | 0.059s   |  **0.055s**  |  0.209s |
> |Avg. Inference |  2.009s  |  1.554s  |  **1.002s**  |
>
> FID (Iteration 20) | Ours | DSB | DSBM-IPF |
> |---|---|---|---|
> |   | 135.4  | N/A | N/A  |
> | Backward model|  N/A |  93.65  |  56.32 |
> | Forward model | N/A | 144  |  98.89 |
>
> **Overall, in this particular experiment, we see that our method makes a trade-off between a small reduction in sample quality for a significant speed-up in training, while also preserving the time-symmetry of the solution.**
>
> We have also tested the same experiment using the CelebA dataset and observed issues with mode collapse with one of the alternative methods. Given the short discussion window and to be fair to the method in question, we have omitted these results but would like to run more ablation tests to possibly include these for the camera-ready version of our paper.
>
> **We believe that your review has now been fully addressed with the new additions and ask that you please consider raising your score.** We are thankful for the suggestions and comments made in your review, and we believe our current version of the paper has been much improved by addressing them!

---

> ### Author Response · Authors · 2024-11-25
>
> **We hope that your review has been fully addressed with the latest PDF update and the previous responses** we provided in the official comments. **We ask that you please refer to these and consider raising your score.** We greatly appreciate the improvements made to our paper thanks to your review.
>
> In specific, we have addressed your concerns by providing a convergence rate for our algorithm, clarifying the difference between our method and SGMs, and including additional experiments such as a comparison to DSB and DSBM for image resampling (see Appendix D on pages 17-19). For more details on modifications made to address your initial review, please refer to the previous official comments in this thread.  If any further comments or questions arise please let us know as soon as possible as the review period is coming to an end.

---

> ### Author Response · Authors · 2024-11-28
>
> As a reminder, we posted an **updated version of the PDF with several additions that we believe fully address the concerns and questions in your original review.** **We kindly ask that you please raise your score in light of these.** If you have any further questions, please let us know as soon as possible. We appreciate your feedback and are eager to respond if further clarification is needed. Your original review had three concerns; here’s a summary of how we addressed all of them in the PDF and previous comments:
>
> * **Convergence rate**: We derived the convergence rate for our algorithm and added it to the proof and statement of Theorem 1 in Section 4.2 Lines 287-290. In brief, the convergence rate is o(1/n), where n is the number of iterates.
>
> * **Comparison to baseline methods**: We added an experiment in Appendix D4 Lines 959-1017 comparing our method to both DSB and DSBM for image resampling. We observe that our method makes a trade-off between a small reduction in sample quality for a significant speed-up in training, while also preserving the time-symmetry of the solution.
>
> * **Clarifying connection to score-based approaches**: We have added a discussion on the connection and difference between MSB and score-based generative modeling (SGM) in Section 2 Lines 109-113. There, we point out that unlike SGMs, our method provides a tool to flow an existing sample somewhere else in the same data distribution with control over the spread of the newly obtained sample. In contrast, SGMs flow samples from a Gaussian to the data distribution.
>
> In addition to modifications directly addressing your review, **we have also added several other experiments to expand the empirical evaluation of our method**, which can be found in Appendix D (pages 17-19).

---

> > ### Comment · Reviewer_vpEm · 2024-12-02
> >
> > Thank you for your clarification and efforts, I will maintain my score.

---

### Author Response · Authors · 2024-11-24

Dear Reviewers,

We would like to check in with all of you as we have not received any responses during the discussion period so far and it’s coming to an end. **We hope that your reviews have been fully addressed with the new PDF update, as well as responses provided in the official comments, and ask that you please consider raising your score.** Should any further clarification or discussion be needed, please let us know as soon as possible. We appreciate all the suggestions and comments in your reviews and believe that our submission has benefited from addressing them. We hope you will consider updating your score in light of these.

For convenience, here is an overall summary of the modifications:
* **New results or clarifications to address the theoretical questions**, e.g. derivation of the convergence rate for our algorithm, clarification of choice of prior. These have been reflected with additions throughout the text (labeled “NEW” in Green or “FIX” in Orange).
* **New experiments**, which can be found in Appendix D (labeled “NEW” in Green) and substantially expand on the empirical evaluation of our method. These new experiments include a **comparison to alternative methods for image resampling, new evaluation metrics, results on the regularity of sample paths and control over proximity**.
For further details on how these address questions specific to your review, please refer to the individual official comments we have made so far.

Sincerely,

The Authors

---

### Author Response · Authors · 2024-12-04
**Final remarks to reviewers**

We thank the reviewers for their helpful comments and suggestions. As the discussion period comes to an end, we outline the changes made to our original submission and present final remarks to support the consideration of our work:

### ***New Experiments (in Appendix D)***

*Comparison to alternative methods.* We added an experiment comparing our method to both DSB and DSBM for image resampling. We observe that our method provides a significant speed-up in training, while also preserving the time-symmetry of the solution.

*Empirical Notion of Distance.* We added an experiment that shows an empirical measure of distances, as suggested by reviewer gCiH, and demonstrates how larger sigma values correspond to quantitatively farther samples. This new set of results demonstrates that our method provides control over sample distance from input, as claimed in theory.

*Regularity of Sample Paths.* As noted by reviewer yMTk, our method “provides optimality with respect to relative entropy, which is a nice property of the sample paths.” To provide quantitative results that analyze the regularity of sample paths, we added an experiment in which we observe that smaller sigmas produce more regular paths.

*Integrity of Initial Distribution.* We present new results using Chamfer distance as an in-distribution metric for the pushforward samples of 2D datasets and new results using FID scores for image resampling of MNIST dataset to demonstrate integrity of the original distribution.

### ***Other Additions***
We have provided clarifications to address all concerns raised in the initial reviews. A number of these were reflected in the text, including a *proof of rate of convergence for our method* and *discussion on the connection and difference between MSB and score-based generative modeling (SGM)*.

### ***Final Remarks***

Multiple reviewers acknowledged the computational advantages of our method, with reviewer gCiH saying, “[Our] innovation significantly reduces computational costs” and reviewer Rjzk describing this reduction on computational burden as “appealing.” These comments were complemented by the new results presented in Appendix D4, where we show a significant reduction in runtime for image resampling when compared to alternative methods. Regarding novelty, reviewer Rjzk notes that our work is “unlike most of the literature which focuses on the [Schrödinger Bridge] between two distributions,” and reviewer yMTk says we “provide an interesting perspective on the Schrödinger bridge problem.” Reviewer gCiH noted our work’s **“capability to produce high-quality proximal samples that are valuable for tasks like data augmentation and generative modeling.”** We believe that these indicate that our work would be a valuable tool for conditional resampling and a welcome addition to the ICLR program.

Once again, we thank the reviewers for their time and hope that the points highlighted above will be taken into consideration. We appreciate the improvements made to our paper based on their feedback.

Sincerely,

The Authors

---

### Meta-Review · Area_Chair_EFLE · 2024-12-19

**Metareview:**

This work is on the topic of Schrödinger Bridge. The authors propose a new method to solve a special class of Schrödinger Bridge problems where the two marginal distributions coincide. Both theoretical analysis and practical implementation are provided. It is claimed the proposed algorithm can reduce the computational cost of solving this class of Schrödinger bridge problem compared to the standard Sinkhorn algorithm. One major weakness pointed out by the reviewers is the empirical study. The experiments lack high dimensional examples and thorough comparisons to existing methods. A more serious issue is that the proposed method and theoretical result are problematic. In particular, a core discover of this paper summarized in Proposition 3 turns out to be wrong. To see this, one can consider a static discrete Schrödinger Bridge problem where the goal is to find a matrix with given column and row sums and is closest to the prior. In this simple setting, it is easy to see the KKT conditions of these two optimization problems in Proposition 3 are different.

**Additional Comments On Reviewer Discussion:**

The reviewers raise some questions on the results as well as the presentation of the paper. The authors reply by modifying the paper, adding experiments in the paper, and adding clarifications in the response. Some reviewers are not convinced. Overall, the reviewers do not seem to be excited about the results in this paper.

---

> ### Public Comment · ~Leticia_Mattos_Da_Silva1 · 2025-02-05
>
> We thank the reviewers and the AC for pointing this out. We really appreciate the careful review! This is exactly the kind of feedback that makes the peer review process so important. We now recognize the issue with Proposition 3; our proof relied on a shaky proposition stated in past work. We have since developed a better understanding of the problem. We are working on a revised version that corrects this particular issue, and we're excited to modify our approach based on the insight provided and our new understanding. We encourage readers to please stay tuned and check out our revision when it is ready. Thank you again for the valuable review!
>
> Sincerely,
> The Authors.

---

### Decision · Program_Chairs · 2025-01-22

Reject